# TRIM: Scalable 3D Gaussian Diffusion Inference with Temporal and Spatial Trimming

**Zeyuan Yin    Xiaoming Liu**
Department of Computer Science and Engineering,
Michigan State University, East Lansing, MI, USA
{zeyuan, liuxm}@msu.edu

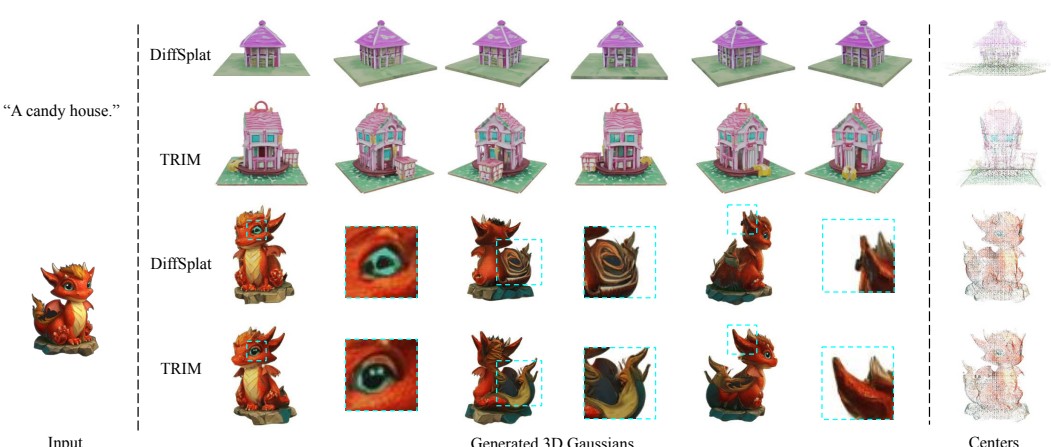

**Figure 1:** Comparison between baseline DiffSplat [1] (Rows 1&3) and our TRIM (Rows 2&4). For text-to-3D, TRIM yields houses more aligned with the "candy" characteristics. For image-to-3D, TRIM generates more realistic details on eyes, tails, and horns. TRIM also reduces inference time from 8 to 5 seconds.

## Abstract

Recent advances in 3D Gaussian diffusion models suffer from time-intensive denoising and post-denoising processing due to the massive number of Gaussian primitives, resulting in slow generation and limited scalability along sampling trajectories. To improve the efficiency of 3D diffusion models, we propose **TRIM** (**T**rajectory **R**eduction and **I**nstance **M**ask denoising), a post-training approach that incorporates both temporal and spatial trimming strategies, to accelerate inference without compromising output quality while supporting the inference-time scaling for Gaussian diffusion models. Instead of scaling denoising trajectories in a costly end-to-end manner, we develop a lightweight selector model to evaluate latent Gaussian primitives derived from multiple sampled noises, enabling early trajectory reduction by selecting candidates with high-quality potential. Furthermore, we introduce instance mask denoising to prune learnable Gaussian primitives by filtering out redundant background regions, reducing inference computation at each denoising step. Extensive experiments and analysis demonstrate that TRIM significantly improves both the efficiency and quality of 3D generation.

39th Conference on Neural Information Processing Systems (NeurIPS 2025).

# 1 Introduction

Recent advancements [2, 3, 1] in text-to-3D generation have transformed creative industries, enabling the synthesis of high-fidelity 3D objects from textual descriptions for applications in filmmaking, gaming design, and virtual reality. Inspired by diffusion models' success in 2D image generation [4, 5], text-to-3D generation has made significant progress with the integration of diffusion models, enabling high-quality 3D synthesis from text prompts. Recent works, such as DiffSplat [1] and GaussianAtlas [6], have shown that re-purposing image diffusion models with a 2D prior for Gaussian primitives generation can produce highly realistic and semantically consistent 3D objects.

To better utilize generative models, numerous post-training techniques have been proposed to enhance models' efficiency and output quality. In the 2D domain, image generation speed has been effectively increased via efficient inference methods like diffusion distillation [7, 8] and model compression [9, 10]. Concurrently, generation quality is often improved via post-training strategies such as reinforcement learning (RL)-based fine-tuning [11, 12] and best-of-N selection with inference-time scaling [13, 14]. However, migrating these 2D diffusion post-training techniques to 3D diffusion is widely regarded as difficult due to two main restrictions. First, the unstructured nature of the 3D Gaussian Splatting representation, where numerous primitives are scattered in 3D space, complicates structured compression and optimization. The second is the extensive computational pipeline where the 3DGS diffusion model sequentially executes image-to-3D reconstruction, 3DGS generation, and rendering, shortly as a Recon-Gen-Render process. Compared to simple 2D denoising, this three-stage process incurs considerably more computation cost and hinders its inference time scaling ability and further RL-based fine-tuning.

To address these challenges, we propose **TRIM** (**T**rajectory **R**eduction and **I**nstance **M**ask denoising), a novel post-training framework designed for the 3D diffusion's Recon-Gen-Render pipeline to enhance the inference efficiency and scalability without compromising quality. Our design is motivated by identifying and resolving two key inefficiencies in existing 3D diffusion pipelines. (1) At the trajectory level, we observe that inference-time scaling by increasing the number of sampled trajectories significantly improves the chance of producing high-quality 3D assets, but requires extensive computation in 3D diffusion models. To overcome this costly end-to-end denoising, our TRIM introduces a latent selector to identify promising latent trajectories early in the denoising process. This significantly reduces the number of fully denoised trajectories and 3D renderer calls. (2) At the token level, we identify a key drawback of prior works: the unnecessary optimization of transparent background regions, which leads to inefficient denoising. To resolve this, TRIM develops an instance masking mechanism to detect and eliminate background splat tokens. This focuses computational resources on the foreground object. Subsequently, a post-denoising correction module utilizes the instance mask to mitigate rendering artifacts by adjusting Gaussian primitive parameters.

Our TRIM conducts the temporal and spatial trimming to address both trajectory-level and token-level redundancies, achieving substantial savings in computational resources. Our main contributions are summarized as follows:

- We propose a novel framework for accelerating 3D Gaussian diffusion inference by performing trajectory-level and token-level pruning, which involves a three-stage inference procedure of trajectory reduction, instance mask denoising, and post-denoising correction.

- Our post-training framework is model-agnostic and can be integrated into a variety of transformer-based 3D diffusion model backbones without the need of retraining.

- Extensive experiments demonstrate that TRIM achieves higher efficiency in text-to-3D and image-to-3D generation and improves generation quality across multiple benchmarks.

# 2 Related Work

**3D Generation.** Diffusion models [15] have become a dominant approach for the visual generative task. There are a lot works [2, 16, 17, 1, 18] proposed for 3D object generation in the 3D domain. Specifically, GVGEN [2] achieves 3D Gaussian diffusion by transforming sparsely located 3D Gaussians into more structured 3D volumes. LRM [16] is a large reconstruction model by scaling the training data and parameters in feed-forward transformer-based models. InstantMesh [17] further integrates novel view synthesis models into LRM with more reference images from different

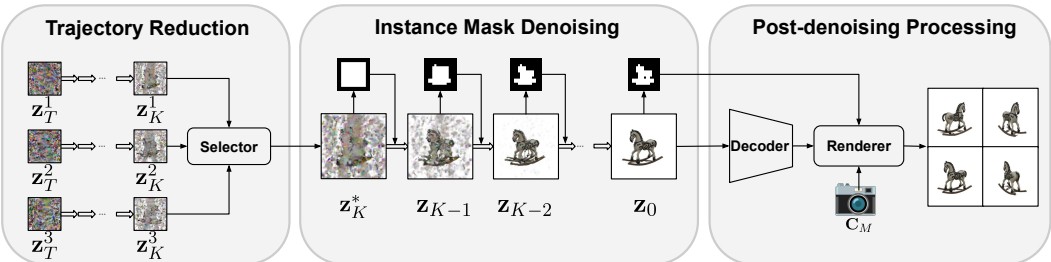

**Figure 2:** Overview of our TRIM framework. It consists of three stages: given a text prompt *A rocking horse with scroll-work*, in the first stage, multiple denoising trajectory candidates are reduced to one trajectory with high-quality potential. In the second stage, an instance mask is performed to simplify background regions during denoising process. In the last stage, the Gaussian primitive parameters are corrected by the mask.

viewpoints. To leverage the prior knowledge in the advanced image diffusion models, like PixArt-$\Sigma$ [4] and Stable Diffusion 3 [5], which are trained on the large-scale Internet data, DiffSplat [1] and Gaussian Atlas [6] transform the 2D model structure for 3D Gaussian representation to enable fine-tuning a well-pretrained image diffusion for 3D Gaussian generation.

**Inference-Time Scaling.** Inference-time scaling has been demonstrated to be an effective strategy for improving performance in Large Language Models (LLMs), enhancing reasoning capabilities and output quality through longer inference [19, 20, 21]. Inspired by LLMs, similar trends are explored and validated in image diffusion models [22, 13, 14], where sampling multiple denoising trajectories or noise initializations improves generation quality. Several recent works emphasize the significance of diverse noise seeds in diffusion sampling, and propose strategies to optimize or select better denoising trajectories through scoring [13], training-free search [14, 23], or optimization mechanisms [24]. However, inference-time scaling remains underexplored in 3D diffusion models. Compared to image generation, 3D generation models, like DiffSplat [1], involve an additional rendering process and higher denoising costs due to a large amount of Gaussian primitives, making it impractical to directly apply evaluation or optimization on noisy 3D assets. In this work, we address this gap by exploring inference-time scaling in 3D diffusion via a lightweight trajectory selection framework that enables efficient quality improvement with minimal overhead.

**Efficient Diffusion Models.** Recent works have explored various strategies to improve the efficiency of diffusion-based generative models. One prominent direction is to reduce the token complexity in transformer-based architectures. SANA [10] introduces a linear attention mechanism tailored for high-resolution synthesis, enabling linear-time complexity with respect to sequence length. Token Merging (ToMe) [25] proposes to merge redundant tokens with similar representation, significantly accelerating diffusion inference with minimal quality loss. Similarly, DiffCR [26] learns adaptive token compression ratios per layer and timestep, providing a fine-grained trade-off between speed and quality. However, these methods often rely on architecture updates and model retraining, which limit their applicability in 3D domains. In contrast, we propose a training-free instance mask denoising method to eliminate redundant background tokens during inference, which is easily plugged into the off-the-shelf transformer-based 3D diffusion models without any retraining cost.

## 3 Method

In this section, we present the proposed TRIM framework. First, we provide an overview of the method in Sec. 3.1. Then, we detail two main components integrated into diffusion inference: (1) Trajectory Reduction in Sec. 3.2 and (2) Instance Mask Denoising in Sec. 3.3.

### 3.1 Overview of TRIM

Our proposed TRIM framework accelerates diffusion inference for 3D generation by integrating two novel components: (1) **Trajectory Reduction**, which uses a latent selector to identify and retain the most promising trajectory at an early timestep, thereby reducing redundant denoising on multiple trajectories; and (2) **Instance Mask Denoising**, which leverages a training-free mechanism to detect and progressively mask background regions to reduce token computation in denoising transformers.

As illustrated in Figure 2, TRIM initially samples diverse noises and generates multiple denoising trajectories. A lightweight *Latent Selector* is trained to evaluate intermediate latents and identify the trajectory with high-quality potential. The chosen trajectory continues for the remaining denoising steps while other trajectories are terminated, significantly reducing inference cost. In parallel, TRIM introduces a spatial mask strategy to compress background tokens during denoising. This mask is self-detected from latent features using a reference-attention mechanism and progressively expanded to ensure stability. Integrated with token merging and post-denoising correction, TRIM enables efficient background filtering while preserving generation quality. These two components are integrated into the diffusion inference for faster and higher-quality 3D generation.

## 3.2  Trajectory Reduction

To enable early trajectory reduction, we introduce a latent selector to identify the candidate with the highest quality potential during the denoising process in the latent space. We formulate training for the latent selector as a knowledge distillation problem. The goal is to distill knowledge from the Decoder-Renderer-Evaluator joint model, where the selector learns to predict the quality of latent representations by approximating the relationship between latent splats on the denoising trajectory and scores evaluated at the final rendered images.

Given a prompt $p$ and a denoising latent trajectory, denoted as $\mathcal{T} = \{\mathbf{z}_T, \mathbf{z}_{T-1}, \cdots, \mathbf{z}_0\}$, the latent selector is optimized by aligning its prediction on an intermediate latent $\mathbf{z}_t$ with the evaluation score of the Evaluator on the rendered images derived from $\mathbf{z}_0$. Specifically, the objective is formulated as:

$$\theta_{\text{selector}} = \arg\min_\theta \mathcal{L}\left(\text{Selector}_\theta(p, \mathbf{z}_t), \text{Evaluator}\left(p, \text{Renderer}\left(\mathbf{C_M}, \text{Decoder}(\mathbf{z}_0)\right)\right)\right), \quad (1)$$

where $\mathcal{L}(\cdot)$ denotes a distillation loss function and $\mathbf{C_M}$ represents the camera matrix used for Gaussian splatting rendering. The latent $\mathbf{z}_t$ is selected at timestep $t$ from the denoising trajectory $\mathcal{T}$.

Instead of directly training the selector by distilling knowledge from a 3D diffusion model, we adopt an offline distillation strategy, consisting of two decoupled stages: data synthesis and selector training, to reduce training costs significantly. In the first stage, we infer the text-to-3D diffusion model to generate 3D views and apply metric-based evaluations, constructing a dataset of {trajectory, score} pairs. In the second stage, we train the latent selector on this pre-processed dataset for pairwise selection tasks, predicting the trajectory with higher quality. This offline distillation strategy effectively alleviates the difficulty in training the latent selector.

**Data Synthesis.** As shown in Algorithm 1 in the Appendices, we construct a triplet dataset of {description prompts, denoising trajectories, evaluation scores}. Specifically, we leverage ChatGPT to generate a set of 100 vivid textual prompts, denoted as $\mathcal{P} \in \mathbb{R}^{100 \times L_{\text{prompt}}}$, each describing a single object, multiple objects, or an object within a scene. For each prompt $p_j \in \mathcal{P}$, we generate 64 diverse latent trajectories using a denoising model $\mathcal{G}$. Each trajectory, denoted $\mathcal{T}_i = \{\mathbf{z}_T^i, \mathbf{z}_{T-1}^i, \ldots, \mathbf{z}_0^i\}$, where $\mathbf{z}_t^i$ is the latent state at timestep $t$, is produced by inferring $\mathcal{G}$ with the random seed $r_i$. The final latent state $\mathbf{z}_0^i$ of each trajectory is decoded into a 3D object representation $\mathcal{O}_i$ using a VAE decoder and then this object is rendered into a set of images $\mathcal{I}_i \in \mathbb{R}^{V_{\text{out}} \times H \times W}$ from $V_{\text{out}}$ viewpoints, specified by a camera matrix $\mathbf{C}_M$. The quality of the rendered images is evaluated using an off-the-shelf image metric evaluator, yielding a scalar score $s_i$. This evaluation score $s_i \in \mathbb{R}$ serves as the proxy for the overall quality of the whole denoising trajectory $\mathcal{T}_i$ that produced $\mathbf{z}_0^i$.

Finally, we assemble a triplet dataset $\mathcal{D} = \left\{\left(p_j, \{(\mathcal{T}_i, s_i)\}_{i=1}^M\right)\right\}_{j=1}^N$, where $N$ is the number of prompts and $M$ is the number of random seeds ($N = 100$, $M = 64$ in our settings). This generation approach ensures diversity in the generated outputs by leveraging stochasticity introduced by the distinct seeds, enabling an expansive exploration space for selector training.

**Latent Selector Training.** Considering challenges in distilling knowledge from a large Decoder-Renderer-Evaluator joint model to a lightweight selector in Eq. 1, we turn to train the latent selector under a pairwise selection setting. Given two intermediate latent codes $\mathbf{z}_t^1$ and $\mathbf{z}_t^2$ at an intermediate timestep $t$ $(0 < t < T)$ along two denoising trajectories $\mathcal{T}_1$ and $\mathcal{T}_2$ conditioned on the same prompt, the model is trained to predict which latent corresponds to a higher-quality 3D object. The supervision signal is determined based on the associated evaluation scores, with the latent corresponding to the higher score designated as the preferred choice.

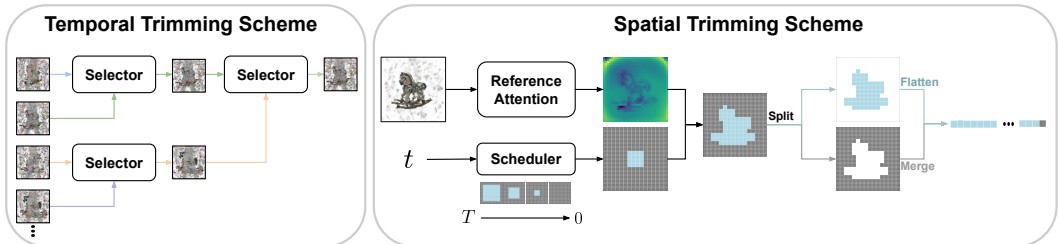

**Figure 3:** Details about temporal (left) and spatial (right) trimming schemes. In temporal trimming, high-quality trajectories are selected early using a lightweight selector. In spatial trimming, the mask is detected and utilized to separate and merge background tokens, reducing the number of tokens processed during denoising.

Our lightweight selector architecture comprises a CNN feature extractor and an MLP discriminator with a prompt fusion mechanism. Specifically, given two latent representations $\mathbf{z}_t^1$ and $\mathbf{z}_t^2$ with scores $s_1$ and $s_2$, the selector is trained to predict the preference label $y = \mathbb{1}(s_1 > s_2)$, which is 1 if the evaluation score $s_1$ exceeds $s_2$ and 0 otherwise. Firstly, we extract the latent features $\mathbf{f}_1 = \mathrm{CNN}(\mathbf{z}_t^1)$ and $\mathbf{f}_2 = \mathrm{CNN}(\mathbf{z}_t^2)$. The MLP discriminator, equipped with a prompt fusion mechanism, concatenates the feature difference $\mathbf{f}_1 - \mathbf{f}_2$ with the prompt embedding $\mathbf{e}_p$ (from prompt $p$), then predicts the label $\hat{y} = \mathrm{MLP}([\mathbf{f}_1 - \mathbf{f}_2; \mathbf{e}_p])$. The selector (CNN + MLP) is optimized using a binary cross-entropy loss:

$$\mathcal{L}_{\mathrm{BCE}}(y, \hat{y}) = -\left(y \log \sigma(\hat{y}) + (1 - y) \log(1 - \sigma(\hat{y}))\right), \tag{2}$$

where $\sigma(\cdot)$ denotes the sigmoid activation function.

**Temporal Trimming Scheme.** In the diffusion model inference with $N$ samples, we leverage the well-trained selector to reduce denoising trajectories at a given $t$ timestep using a pairwise tournament selection strategy, shown in the left scheme of Figure 3. Identifying a high-quality trajectory early reduces the total denoising steps, originally $NT$. With the reduction strategy, we denoise $N$ candidates from timestep $T$ to $t$, then 1 candidate from $t - 1$ to 0, totaling $NT - (N - 1)t$ steps, saving $(N - 1)t$ steps. Furthermore, since the number of final latents is reduced from $N$ to 1, the post-denoising processing costs, including VAE decoding and Gaussian splatting rendering, are also reduced by $N$ times.

### 3.3 Instance Mask Denoising

We introduce an instance masking mechanism during denoising to eliminate redundant background splats in the Gaussian splatting grid. Specifically, as shown in the right scheme of Figure 3, we first detect a mask to distinguish the instance and background based on the latent representation. Then the mask is gradually applied to drop the background and maintain the instance region in the latent representation, effectively reducing the number of tokens processed in the denoising transformer.

**Instance Mask Detection via Reference Attention.** Inspired by the patch-level attention mechanism in DINO [27], which highlights semantically salient regions via attention between the [CLS] token and patch tokens, we propose a lightweight, model-free instance mask detection via *corner-reference attention* to detect a binary segmentation mask that distinguishes instance regions from the background on the latent representation. Specifically, we observe that the four corners of the latent feature grid typically correspond to transparent background regions. We extract and aggregate features from these corner regions to form a reference token, denoted as [REF]. For each patch in the feature grid, we compute the similarity between its feature and the [REF] token. Regions with low similarity to [REF] are treated as potential instance regions. We threshold the similarity map using a predefined value $\tau$ to obtain the binary mask, where 0 and 1 represent the foreground instance and background regions, respectively. Corner-reference attention is less reliable in early denoising steps due to noisy latent representations, which hinder accurate instance localization. As denoising progresses, the latent becomes more structured, enabling more precise mask detection. Thus, we apply the mask selectively during the middle-to-late denoising stages.

**Progressive Mask Expansion Scheduler.** To enable a smooth transition from unmasked to masked region denoising, we adopt *progressive mask expansion*, a spatially controlled strategy that gradually expands the masked area from the outer boundary toward the center of the latent grid. As illustrated in the scheduler module of Figure 3, assuming a $16 \times 16$ latent patch grid (*e.g.,* in Stable Diffusion 3),

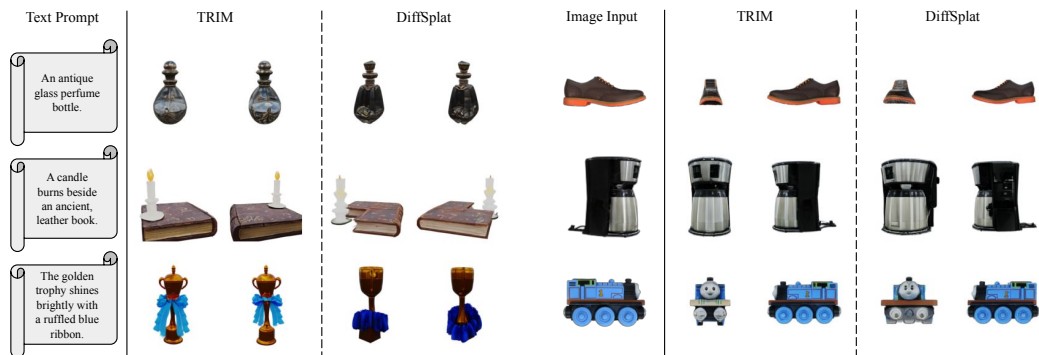

**Figure 4:** Qualitative results and comparisons on Text-to-3D (left) and Image-to-3D (right) generation.

we divide the denoising process into four phases. In the first phase, only the outermost 2 row/column patches are subjected to background masking. In subsequent phases, the masked region is expanded to include 4, 6, and finally 8 (entire grid) row/column patches. The progressive expansion reduces the risk of artifacts introduced by hard masking and improves robustness during early-to-mid denoising steps, while still enabling efficient background reduction in later stages.

**Token Merging and Padding.** Once the instance mask is obtained via corner-reference attention and the mask region is scheduled through progressive mask expansion, they are combined to produce a final binary mask that separates foreground instance and background regions. All background-region tokens in the latent grid are aggregated into a merged background token, denoted as [BG], and then concatenated to the flattened foreground instance token sequence. The instance-region tokens along with [BG] token are fed into the denoising transformer. After denoising, the output [BG] token is padded back in the masked background positions, restoring the full 2D latent grid structure. The token merging and padding process ensures compatibility with downstream modules, including VAE decoding and the Gaussian splatting rendering.

**Post-denoising Correction.** After denoising, the latent representation is decoded into a set of Gaussian primitives with explicit parameters, which are then rendered into images using Gaussian splatting guided by a camera matrix $\mathbf{C}_M$. However, since our approach is training-free and directly employs an off-the-shelf Gaussian diffusion model for inference, the model lacks prior knowledge to the introduced mask and [BG] token during training. Consequently, the [BG] token is not optimally denoised to a fully transparent background, leading to artifacts in the rendered images. To address this, we utilize the mask generated from the last step's latent representation to correct the parameters of Gaussian primitives, as shown in Phase 3 of Figure 2. Specifically, we set the background-region Gaussian primitives' opacity values to zero. The corrected parameter effectively eliminates background artifacts and the influence of background primitives on the rendered images.

## 4  Experiments

### 4.1  Experimental Settings

We adopt DiffSplat [1] as our main backbone model, trained on the G-Objaverse dataset [28]. For the text-to-3D generation task, we evaluate on T³Bench [29], which consists of 300 descriptive prompts about a single object, a single object with surrounding context, or multiple objects. We report the CLIP Similarity Score [30] and CLIP R-Precision [31] using ViT-B/32 to measure alignment between the input prompts and the rendered images. Additionally, we utilize ImageReward [32] to assess perceptual quality based on human aesthetic preference. For the image-to-3D generation task, we randomly select 300 objects from the Google Scanned Objects (GSO) dataset [33]. For each object, the front-facing image serves as input for 3D generation, while rendered images from other viewpoints act as ground-truth. We evaluate reconstruction fidelity using PSNR, SSIM, and LPIPS [34]. For our TRIM configuration, trajectory reduction is performed at the midpoint of the denoising process ($t = T/2$), and the selector model is trained on latent representations extracted at this intermediate timestep. More details about experiments are contained in the Appendices.

**Table 1:** Quantitive results on T³Bench for text-to-3D generation.

| Benchmark | Metric | GVGEN [2] | DIRECT-3D [3] | LGM [35] | DiffSplat [1] | **TRIM (Ours)** |
|---|---|---|---|---|---|---|
| Single Object | CLIP Sim.$_\%$ | 23.66 | 24.80 | 29.96 | 30.95 | **31.58** |
| | CLIP R-Prec.$_\%$ | 23.25 | 30.75 | 78.00 | 81.00 | **81.42** |
| | ImageReward | -2.15 | -2.00 | -0.72 | -0.49 | **0.12** |
| Single Object w/ Sur. | CLIP Sim.$_\%$ | 22.65 | 23.05 | 27.79 | 30.20 | **31.48** |
| | CLIP R-Prec.$_\%$ | 26.75 | 25.75 | 55.00 | 80.75 | **88.25** |
| | ImageReward | -2.25 | -2.19 | -1.77 | -0.67 | **-0.50** |
| Multiple Objects | CLIP Sim.$_\%$ | 21.48 | 21.89 | 27.07 | 29.46 | **30.11** |
| | CLIP R-Prec.$_\%$ | 8.00 | 7.75 | 51.00 | 69.50 | **70.01** |
| | ImageReward | -2.27 | -2.24 | -1.73 | -0.84 | **-0.24** |

**Table 2:** Quantitive results on GSO dataset for image-to-3D reconstruction.

| Metric | LGM [35] | InstantMesh [17] | DiffSplat [1] | TRIM (Ours) |
|---|---|---|---|---|
| PSNR↑ | 14.90 | 15.53 | 16.20 | **16.78** |
| SSIM↑ | 0.71 | 0.77 | 0.79 | **0.82** |
| LPIPS↓ | 0.25 | 0.22 | 0.19 | **0.17** |

## 4.2 Text-to-3D Generation

As shown in Table 1, on the *Single Object* category of T³Bench, TRIM achieves the highest CLIP Similarity and CLIP R-Precision, indicating stronger semantic alignment between the generated 3D assets and the input text prompts. Moreover, TRIM is the only method to yield a positive ImageReward score of 0.12, suggesting higher visual fidelity from a human preference perspective. On more challenging cases such as *Single Object with Surroundings* and *Multiple Objects*, TRIM maintains a clear lead over all the baseline methods, with improvements of 0.73 and 0.65 in CLIP Similarity over DiffSplat, respectively. The ImageReward scores in these sub remain competitive, highlighting TRIM's robustness in complex scenes. These results demonstrate the effectiveness of TRIM's temporal and spatial trimming strategies on enhancing the quality of generated 3D assets.

## 4.3 Image-to-3D Reconstruction

Table 2 and Figure 4 report the quantitative and qualitative results on the GSO dataset for the image-to-3D reconstruction task. TRIM improves the reconstruction performance over strong baselines such as DiffSplat, but the gains are relatively smaller than those observed in the text-to-3D setting. We attribute this to the stronger conditioning signal provided by the input images compared to text prompts, which inherently reduces the diversity among different sampled trajectories. As a result, the benefit from inference-time trajectory scaling becomes less obvious. Nevertheless, TRIM consistently enhances the PSNR, SSIM, and LPIPS scores, demonstrating its general effectiveness in improving 3D reconstruction quality under image conditioning.

## 4.4 Ablation and Analysis

**Qualitative Analysis of Inference Step Scaling.** Figure 5 presents qualitative comparisons of 3D assets generated by DiffSplat with an SD-3.5-Medium backbone using different numbers of inference steps $T = [10, 40, 80]$. As the number of steps increases, the generated outputs generally become more refined and detailed. However, beyond a certain step count, the visual improvements begin to saturate or even degrade slightly. In the first example about single object generation, excessive denoising introduces fake details or artifacts, such as debris-like details, which aren't implied by the prompt. In the second example about multiple object generation, longer denoising leads to semantic inconsistencies: the box handle becomes unnaturally large, and the number of pocket watches conflicts with the prompt description. In the third example about the single object with surroundings, the model fails to preserve semantic details such as the human figure at the higher inference steps. This phenomenon, where more inference steps may cause semantic drift or over-smoothing, particularly for complex scenes, has been observed in recent studies [36, 37, 13]. These observations highlight the

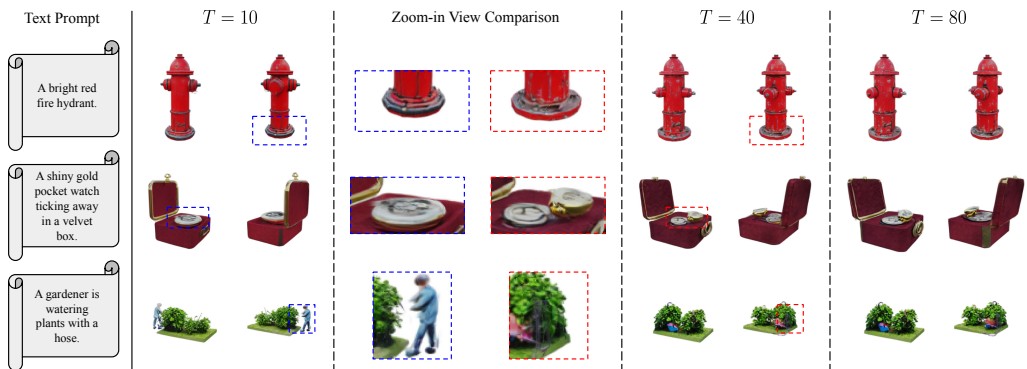

**Figure 5:** Qualitative comparisons on inference step scaling. Best viewed with zoom.

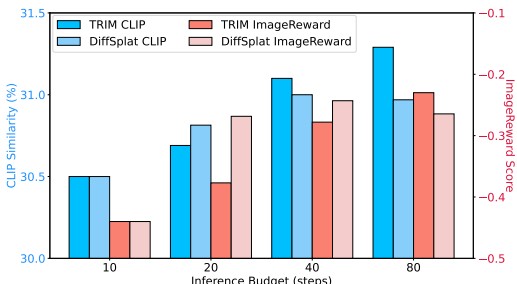

**Figure 6:** Trajectory scaling vs. inference step scaling. TRIM improves steadily via trajectory scaling, while DiffSplat plateaus and eventually degrades with step scaling.

**Figure 7:** Ablation results on selector application time. Each data point indicates the performance when applying the selector at a certain timestep.

limitation of naive inference-step scaling and motivate the need for smarter inference-time trajectory scaling strategies.

**Analysis of Inference-time Scaling.** Figure 6 compares the evaluation results of Trajectory scaling and inference step scaling. To ensure a fair comparison under the same total number of inference steps, the baseline method DiffSplat scales the number of denoising steps from 10 to 80 on one trajectory, while TRIM fixes the denoising step count to 10 and scales the number of sampled trajectories from 1 to 8. The results reveal that TRIM with trajectory scaling achieves steadily higher performance across both CLIP Similarity and ImageReward scores as more trajectories are sampled. In contrast, the DiffSplat with the inference step scaling shows limited improvement in CLIP scores and a clear decline in ImageReward scores with more denoising steps, likely due to over-smoothing or the accumulation of artifacts. These findings suggest that inference-time computation is better spent on trajectory diversity rather than excessively long denoising steps.

**Ablation on Selector Architecture.** We explore the design space of the selector model using a C-layer CNN as a feature extractor followed by an L-layer MLP as the discriminator, optionally conditioned on the text prompt from T³Bench-Single. To evaluate selector performance, we report two kinds of results on selector training and 3D generation: (1) *pairwise accuracy* measures binary classification accuracy on the validation set of pairwise comparisons during selector training; (2) *CLIP similarity* and *ImageReward* scores measure the generation quality of output 3D objects after applying the selector to trajectory reduction and rendering. Table 3 shows that a convolutional feature extractor is crucial for selector performance. Specifically, adding one convolutional layer, from *FC2* to *Conv1-FC2*, leads to a substantial improvement in both pairwise accuracy of $20.82\%$ and CLIP similarity of $0.75\%$. We further notice that increased model complexity does not yield additional gains for the pairwise comparison task. This suggests that a lightweight selector with a single convolutional layer and a two-layer MLP is sufficient to capture discriminative latent features for effective selection. Moreover, its minimal computational overhead ensures that it does not introduce

**Table 3:** Ablation results on Selector architecture.

| Architecture | Pairwise Acc.$_\%$ ↑ | CLIP Sim.$_\%$ ↑ | ImageReward Score↑ |
|---|---|---|---|
| w/o selector | – | 30.44 | -0.51 |
| FC2 | 53.36 | 30.83 | -0.40 |
| Conv1-FC2 | **74.18** | **31.58** | **-0.16** |
| Conv1-FC3 | 73.90 | 31.54 | -0.20 |
| Conv2-FC2 | 73.09 | 31.31 | -0.22 |
| Conv1-FC2-Prompt | 73.73 | 31.14 | -0.26 |
| Conv1-FC3-Prompt | 72.91 | 31.09 | -0.28 |

**Table 4:** Computation resource comparison.

| Method | FLOPs (T)↓ | Mem. (GB)↓ | Throughput (step/s)↑ |
|---|---|---|---|
| SD-3.5-Medium | 195.68 | 33.26 | 13.18 |
| + IM | 165.60 | 32.85 | 18.09 |
| + TR | 110.07 | 33.55 | 13.18 |
| + TRIM | 106.31 | 33.13 | 18.09 |

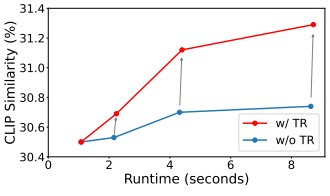

**(a)** Ablation results on TR.

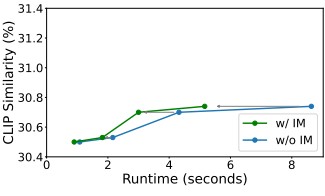

**(b)** Ablation results on IM.

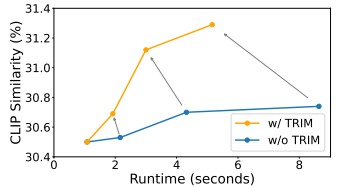

**(c)** Ablation studies on TRIM.

**Figure 8:** Ablation results on trajectory scaling with or without TRIM. The integration of TRIM improves generation quality under reduced computational budgets.

noticeable latency to the diffusion inference process, making it a practical and efficient component for trajectory reduction in 3D diffusion inference scaling.

**Ablation on Selector Application Time.** We investigate the impact of applying the trajectory selector at different denoising timesteps. Figure 7 shows that both CLIP similarity and ImageReward scores improve as the selector is applied in later timesteps, but the performance gains plateau when applying after the 50% progress. This reflects a trade-off between efficiency and quality: applying the selector too early yields poor performance due to high noise in the latent features at the early stage, while applying it too late diminishes the efficiency gain of early pruning. Thus, we adopt the midpoint of the denoising process, 50% of the total steps, as the default selector application point in TRIM.

**Ablation on TRIM.** We analyze the contributions of Trajectory Reduction (TR) and Instance Masking (IM) in Figure 8a and Figure 8b, respectively. We observe that TR primarily improves the CLIP similarity score of the generated 3D outputs with a slight increase in processing time, while IM is more effective at reducing overall runtime while maintaining similar CLIP scores. Thus, as shown in Figure 8c, combining both components in TRIM leads to improvements in both generation quality and inference efficiency. Notably, TRIM gains generation quality and denoising efficiency under less inference runtime, benefiting from our proposed early trajectory reduction mechanism. This demonstrates that TRIM not only accelerates inference but also leads to higher-quality final 3D outputs.

**Diversity Analysis.** To investigate the output diversity, we measure the diversity of generated outputs with or without trajectory reduction on the T$^3$Bench-Single dataset. We repeat the generation process 8 and 16 times, and report the evaluation results on both semantic and geometric diversity via various metrics, including CLIP similarity, ImageReward scores, and Chamfer Distance. Specifically, CLIP similarity and ImageReward scores primarily focus on capturing semantic alignment, while Chamfer

**Table 5:** Results on output diversity with and without trajectory reduction.

| Metric | TR | # repeat of 8 | # repeat of 16 |
|---|---|---|---|
| CLIP Sim. | | 30.89 ± 0.21 | 30.95 ± 0.16 |
| | ✓ | 31.51 ± 0.14 | 31.53 ± 0.11 |
| ImageReward | | -0.45 ± 0.05 | -0.48 ± 0.04 |
| | ✓ | 0.12 ± 0.04 | 0.11 ± 0.03 |
| Chamfer Distance | | 1840, [952, 3060] | 1791, [699, 4435] |
| | ✓ | 2042, [1187, 2813] | 2310, [1187, 4079] |

Distance is used for measuring the distance between two point clouds, which is more sensitive to geometric diversity.

In Table 5, we report the average evaluation performance along with the standard deviation on CLIP similarity and ImageReward scores, where the standard deviation is generally regarded as a key indicator for evaluating diversity. Table 5 also presents the averaged Chamfer Distance (non-normalized), along with the minimum and maximum values (avg., [min., max.]), via the pair-wise comparison among all the repeated generation results. The reported maximum value of the Chamfer Distance is important to reflect the maximum pairwise difference among multiple outputs and indicate the upper bound of diversity. For the implementation details about Chamfer Distance, we first extract the locations of Gaussian primitives, converting each generated 3D output into a point cloud, as shown in the rightmost column of Figure 1. We then calculate the Chamfer Distance for every pair of point cloud outputs among the 8 or 16 generated objects. Take the 8-output setting as an example: we calculate the Chamfer Distance for $\binom{8}{2} = 28$ pairs of outputs and report the average, minimum, and maximum values.

For the semantic diversity shown in the Table 5, we observe that the model's diversity is slightly reduced when using trajectory reduction, with an increase in the average performance. We attribute this to the selector's effect with an early discarding of unpromising trajectories. By filtering out the less promising generation trajectories, the selector effectively shifts the output distribution of diversity toward a higher quality range. This means that TRIM sacrifices diversity in the low-quality outputs to improve the average quality of final outputs. For the geometric diversity, we notice that the range of Chamfer Distances is slightly narrowed and the upper bound is consistently lower when using trajectory reduction across both settings. These observations indicate that the trajectory reduction strategy slightly hurts the diversity of outputs and makes the output distribution narrower.

Thus, we can conclude that initial noises primarily drive the diversity of the outputs, and our latent selector acts as a quality filter, enabling the effective selection from a pool of diverse candidates to high-quality outputs with a narrower variance/diversity. These results are well-aligned with our motivation and support our method to prune unpromising trajectories and shift the output distribution towards a high-quality and narrow range.

**Computation Analysis.** We analyze the computational efficiency of our proposed TRIM and present the detailed comparison results in Table 6 of the Appendix. Specifically, we measure the total FLOPs across all denoising steps. Compared to the baseline DiffSplat with the SD-3.5-Medium backbone, Instance Masking (IM) reduces FLOPs by 15.7% and increases throughput from 13.18 to 18.09 steps/s, while also slightly lowering memory usage. In contrast, Trajectory Reduction (TR) reduces the total number of denoising steps in the temporal axis, thus achieving less total FLOPs. Specifically, by selecting a high-quality trajectory at an intermediate timestep $t$, the total denoising cost drops from $NT$ to $NT - (N-1)t$, saving $(N-1)t$ steps. Additionally, as only one trajectory proceeds to post-denoising, the cost of VAE decoding and Gaussian rendering is also reduced by $N$ times. Therefore, TRIM leverages trajectory reduction and instance masking to improve inference efficiency along the temporal and spatial axes, respectively.

# 5 Conclusion

In this paper, we propose TRIM, a post-training framework for accelerating 3D Gaussian diffusion inference through temporal and spatial inference-time trimming. By introducing latent trajectory reduction and instance masking, TRIM effectively reduces low-quality denoising trajectories and redundant background regions while preserving and improving generation quality. Extensive experiments across multiple benchmarks demonstrate that TRIM not only enhances inference efficiency but also enables effective inference-time scaling, outperforming prior state-of-the-art methods in both semantic alignment and aesthetic quality.

**Limitations and Future Work.** Current 3D diffusion pipelines heavily rely on repurposed 2D backbones, leading to inefficient and repetitive 3D-to-2D structural transformations. This limits spatial trimming to the denoising transformer blocks and prevents its application across the full generative pipeline. To overcome this, it is promising to investigate a 3D-structure-aware diffusion that not only leverages the 2D priors but also enables end-to-end spatial trimming for more efficient training and inference.

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

# A  Implementation Details

**Datasets.** For the text-to-3D generation task, T$^3$Bench consists of three groups of prompts: a single object, a single object with surroundings and multiple objects. Each subgroup contains 100 text descriptions. For the image-to-3D generation task, due to the no access to the data selection list and rendering parameters used in baseline methods, we randomly select 300 objects from the GSO dataset. Each object is rendered from four orthogonal viewpoints: front, back, left, and right sides, with a fixed elevation angle of $0°$.

**3D Generation Setting.** Our 3D generation model with the backbone of Stable-Diffusion-3.5-Medium uses the original flow matching Euler ODE solver with 28 steps in the main experiments. The classifier-free guidance scale is set to 7 for text-to-3D generation and 2 for image-to-3D generation. The image input is center cropped and resized to 256×256 resolution.

**Pairwise Data Synthesis.** We first use ChatGPT-4o to generate $N = 100$ text prompts describing diverse objects with various decorations. Following Algorithm 1, we perform data synthesis by sampling $M = 64$ denoising trajectories per prompt using distinct random seeds. For each trajectory, we decode and render the final denoised output $\mathbf{z}_0$ and compute the alignment score $s$ using a CLIP-based evaluator.

For data processing, we construct pairwise training and testing data from the 100 latent-score pairs. Each data point consisted of $(\mathbf{z}_t^1, \mathbf{z}_t^2, s_1 - s_2)$, where $s_1 - s_2$ represents the difference between the scores of the two latent trajectories. The dataset was then split into a 7:3 ratio for training and testing, respectively. A smaller score distance indicates minimal differentiation between two latent trajectories, making accurate distinction challenging but less critical for selection. Conversely, a larger score distance signifies substantial differences, highlighting the importance of correct classification for effective selection. Due to the pairwise combination method, the data distribution is imbalanced, with a disproportionately large portion of data samples exhibiting small score distances. To ensure balanced data training, we group the data into 11 bins based on absolute score distance $|s_1 - s_2|$, with intervals of 0.1, *i.e.*, [0, 0.1), [0.1, 0.2), ..., [1, $\infty$). We ensure that each group contains 200 samples, resulting in a total training dataset of 2200 samples.

**Selector Training Setting.** We employ the AdamW optimizer with a learning rate of 0.001, a weight decay of 0.01, and a cosine weight decay schedule. Training is conducted with a batch size of 64 for 20 epochs. The model takes a pair of latent features $(\mathbf{z}_t^1, \mathbf{z}_t^2)$ as input and predicts the pairwise comparison outcome. The target label is defined as $\mathbb{1}(s_1 > s_2)$, and the loss is computed using binary cross-entropy over the softmaxed prediction. The trained selector achieves over 70% test accuracy on the overall pairwise validation set in Table 3 and exceeds 90% accuracy for test samples with large score gaps ($|s_1 - s_2| > 1$). This indicates that the well-trained selector effectively captures discriminative features in the latent space and is capable of reliably identifying higher-quality trajectories during inference.

---

**Algorithm 1** Data Synthesis for Latent Selector Training

---

1: **Input**: Prompt set $\mathcal{P} = \{p_i\}_{i=0}^N$, # trajectories per prompt $M$, # denoising steps $T$, camera matrix $\mathbf{C}_M$, decoder, renderer, and evaluator models,
2: **Output**: A triplet dataset $\mathcal{D} = \{(p_j, \{(\mathcal{T}_i, s_i)\}_{i=1}^M)\}_{j=1}^N$
3: Initialize empty dataset $\mathcal{D} \leftarrow \emptyset$
4: **for** $j = 1$ to $N$ **do**                                                        ▷ Iterate over prompts
5:     Initialize empty set $\mathcal{S}_j \leftarrow \emptyset$                        ▷ Store trajectory-score pairs for prompt $p_j$
6:     **for** $i = 1$ to $M$ **do**                                                    ▷ Iterate over random seeds
7:         Set random seed $r_i \leftarrow i$
8:         Generate trajectory $\mathcal{T}_i = \{\mathbf{z}_T^i, \mathbf{z}_{T-1}^i, \ldots, \mathbf{z}_0^i\} \leftarrow \mathcal{G}(p_j, r_i)$
9:         Decode final latent $\mathcal{O}_i \leftarrow \text{Decoder}(\mathbf{z}_0^i)$
10:        Render images $\mathcal{I}_i \leftarrow \text{Renderer}(\mathcal{O}_i, \mathbf{C}_M)$
11:        Evaluate quality $s_i \leftarrow \text{Evaluator}(\mathcal{I}_i)$
12:        Add pair to set $\mathcal{S}_j \leftarrow \mathcal{S}_j \cup \{(\mathcal{T}_i, s_i)\}$
13:    **end for**
14:    Add prompt and pairs to dataset $\mathcal{D} \leftarrow \mathcal{D} \cup \{(p_j, \mathcal{S}_j)\}$
15: **end for**
16: **return** $\mathcal{D}$

---

# B  Experiment results

**Computation Cost.** Table 6 presents the ablation study on the computational contributions of the proposed Trajectory Reduction (TR) and Instance Masking (IM) components. We observe that TR significantly reduces the overall FLOPs by decreasing the total number of denoising steps. However, since multiple trajectories are processed in parallel and the runtime is determined by the longest trajectory, the actual GPU memory usage and runtime show a slight increase due to the additional cost introduced by the selector model. In contrast, IM reduces the computational load of the denoising transformer by pruning background tokens, resulting in clear improvements in both throughput and runtime. Thus, the combined TRIM balances temporal and spatial efficiency, leading to improved overall inference performance.

**Table 6:** Computation resource comparison. The best performance values are shown in bold, and the second-best values are underlined. All results are reported based on RTX A6000 GPU.

| Method | FLOPs (T)↓ | Mem. (GB)↓ | Throughput (step/s)↑ | Runtime (second)↓ |
|---|---|---|---|---|
| SD-3.5-Medium | 195.68 | 33.26 | 13.18 | 8.64 |
| + IM | 165.60 | **32.85** | **18.09** | **5.16** |
| + TR | 110.07 | 33.55 | 13.18 | 8.74 |
| + TRIM | **106.31** | 33.13 | **18.09** | 5.24 |

**Results on PixArt backbone.** Our proposed trajectory reduction strategy is designed to be applicable to most diffusion-based architectures, while the instance mask strategy specifically leverages the token structure of Transformer-based backbones. In our main experiments, we choose Stable Diffusion 3.5, the latest diffusion model in the Stable Diffusion series, as the backbone. To further demonstrate generalization, we also apply TRIM to PixArt-Sigma with a different diffusion backbone, and present the results on the $T^3$Bench-Single dataset in the Table 7. These results show that TRIM improves the performance of DiffSplat with the backbone of PixArt-Sigma, demonstrating that TRIM consistently improves both the generation quality and efficiency across various Transformer-based backbones.

**Table 7:** Quantitative results on $T^3$Bench-Single

| Method | CLIP Sim.%↑ | ImageReward Score↑ | FLOPs (T)↓ | Runtime (second)↓ |
|---|---|---|---|---|
| SD-3.5-Medium | 30.95 | -0.49 | 195.68 | 8.64 |
| SD-3.5-Medium + TRIM | 31.58 | 0.12 | 106.31 | 5.24 |
| PixArt-Sigma | 30.73 | -0.30 | 25.18 | 2.76 |
| PixArt-Sigma + TRIM | 31.24 | -0.13 | 14.16 | 2.19 |

**Results on 3DGen-Bench** We apply the CLIP-based 3DGen-Score model in 3DGen-Bench [38] as an additional metric to enhance our evaluation. The 3DGen-Score model requires multi-view RGB images, normal maps, and a text prompt as input. The evaluation output contains five criteria: Geometry Plausibility, Geometry Details, Texture Quality, Geometry-Texture Coherence, and Prompt-Asset Alignment, each with different value scopes. We provide results on the $T^3$Bench-Single dataset in the Table 8, which is evaluated by the 3DGen-Score metric. The 3DGen-Score results show that TRIM achieves better performance than the baseline DiffSplat on all five criteria.

**Table 8:** 3DGen-Bench results on $T^3$Bench-Single

| Method | Geo. Plausibility | Geo. Details | Tex. Quality | Geo.-Tex. | Alignment |
|---|---|---|---|---|---|
| DiffSplat | 6462.91 | 8.14 | 13.43 | 15790.92 | 8544.38 |
| TRIM | 6567.58 | 9.25 | 13.47 | 16144.38 | 8585.02 |

# C  More Visualization

We provide more visualization comparisons on text/image-to-3D generation from DiffSplat and TRIM in this section. Figure 9 shows that TRIM captures finer details, such as the "splattered"

colors around the easel and the "worn-out" characteristics of the tire, demonstrating better alignment with the input text prompts compared to DiffSplat. Figure 10 demonstrates that TRIM produces more structurally consistent 3D shapes with fewer distortions, leading to more realistic and coherent generation.

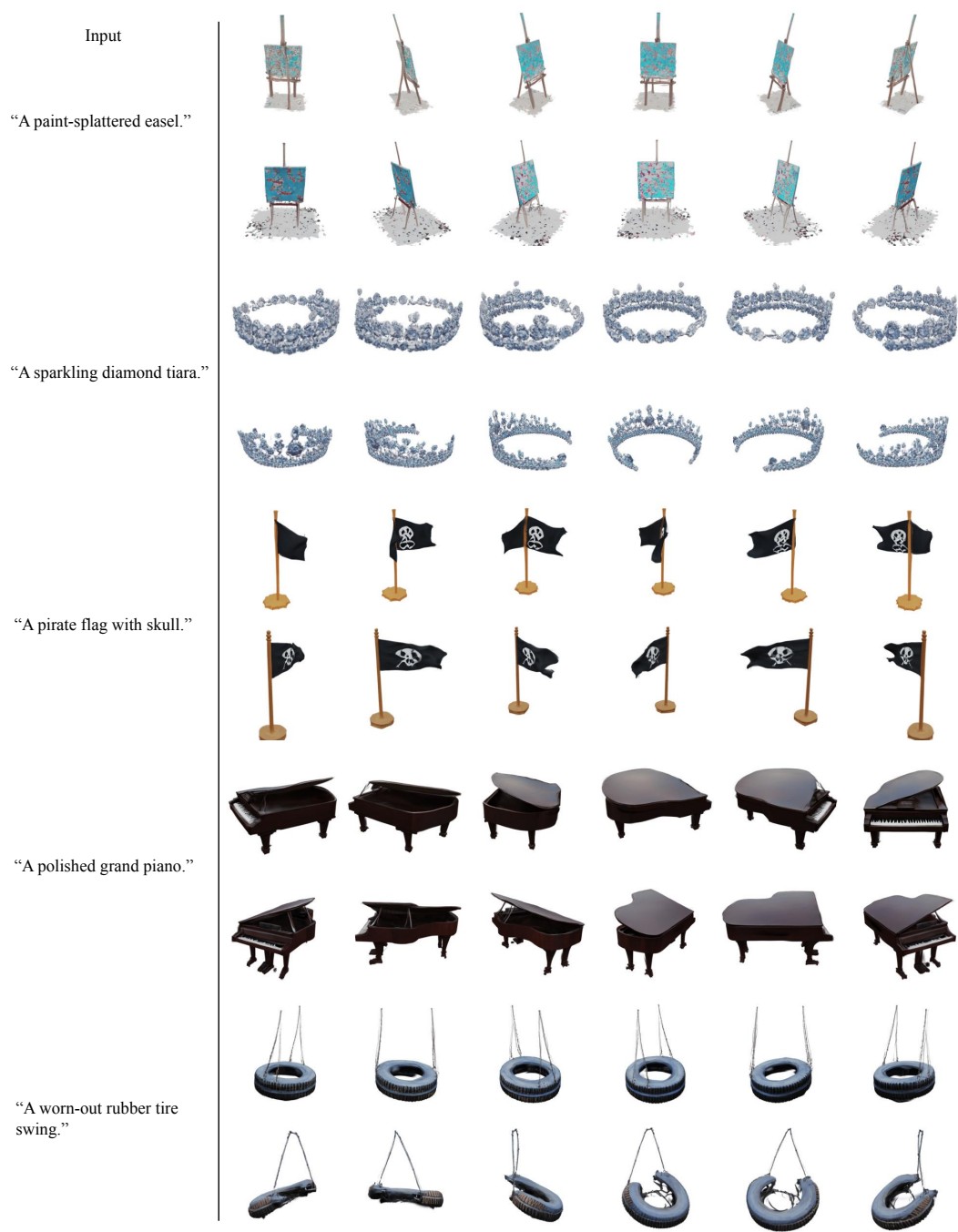

**Figure 9:** Visualization comparisons on T³Bench from DiffSplat (Rows 1,3,5,7&9) and TRIM (Rows 2,4,6,8&10).

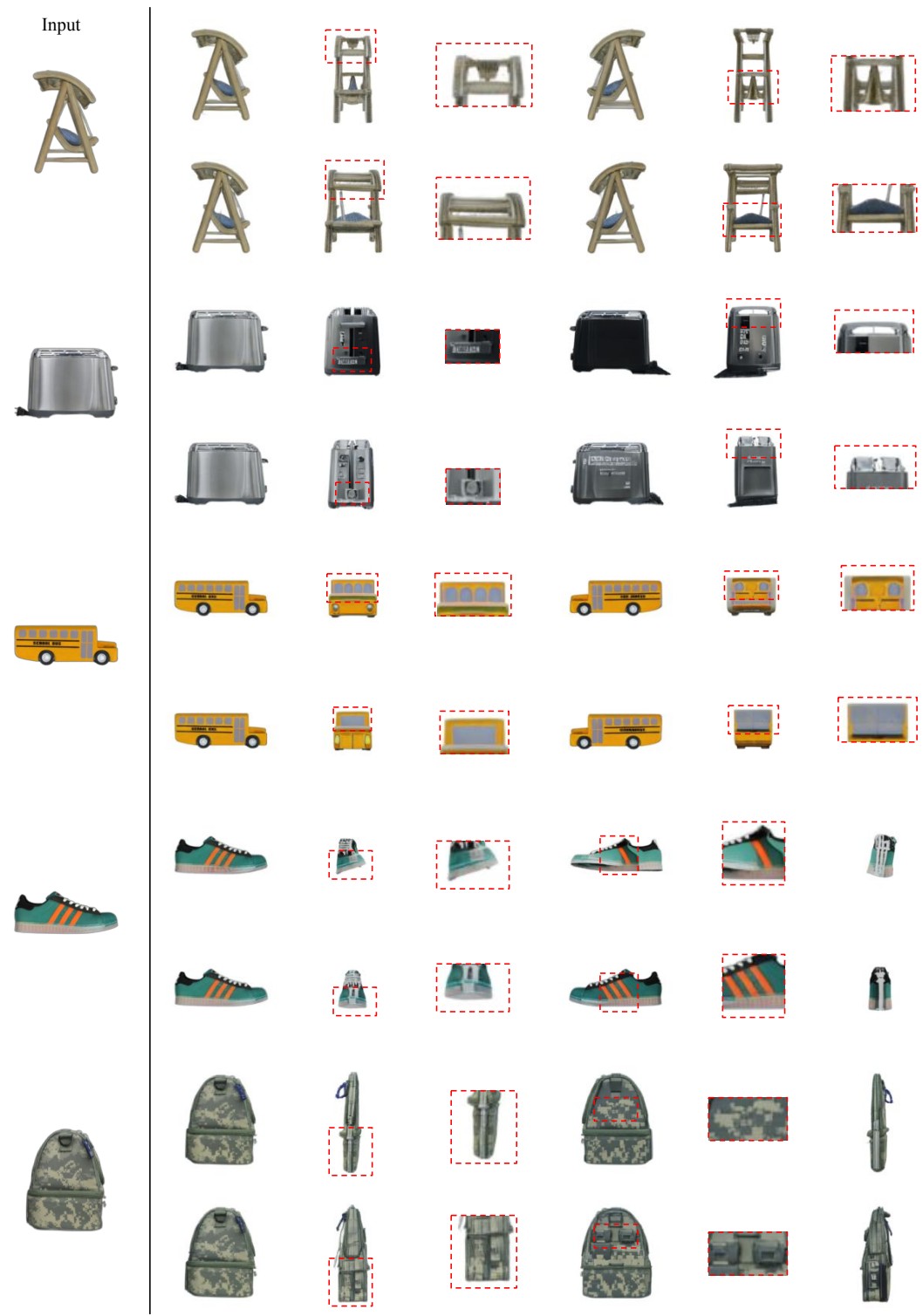

**Figure 10:** Visualization comparisons on GSO dataset from DiffSplat (Rows 1,3,5,7&9) and TRIM (Rows 2,4,6,8&10).

