# OpenReview forum: "TRIM: Scalable 3D Gaussian Diffusion Inference with Temporal and Spatial Trimming"
_NeurIPS.cc/2025/Conference — NeurIPS 2025 poster_

### Official Review · Reviewer_CFPT · 2025-06-21

**Clarity:** 2
**Significance:** 2
**Originality:** 2
**Rating:** 4
**Confidence:** 4

**Summary:**

This paper presents a training-free, model-agnostic framework to enhance the efficiency and quality of 3D diffusion models, through two key components: Trajectory Reduction and Instance Mask Denoising.

Trajectory Reduction involves sampling multiple inference trajectories but fully denoising only those that are likely to yield high-quality results. To enable this, the authors train a latent selector on a synthesized dataset to predict latent quality. This selector approximates the relationship between intermediate latent representations (latent splats) and final rendered image scores.

Instance Mask Denoising focuses on reducing computation by identifying background regions and progressively masking them out during the denoising process, thereby minimizing redundant token processing in the transformer.

The proposed method significantly improves the efficiency of text-to-3D and image-to-3D generation while maintaining or enhancing output quality.

**Questions:**

* Q1: Some baseline results (e.g., DiffSplat) differ notably from those in the original papers, without explanation. Could the authors clarify how these baselines were evaluated or re-implemented?

* Q2: The description of the latent selector’s training is confusing—why is the initially described approach not used, and how is timestep
$t$ sampled during training?

**Ethical Concerns:**

["NO or VERY MINOR ethics concerns only"]

**Final Justification:**

The rebuttal has address most of my concerns except for the diversity problem.
If the authors include the discussion of this problem to the revision version of the paper, I think it is okay to have a score of 4 (borderline accept)

**Limitations:**

yes

**Paper Formatting Concerns:**

I do not see any formatting issue here.

**Quality:**

2

**Strengths And Weaknesses:**

## Strengths
* **S1:** The motivation to discard low-quality samples early in the diffusion process is well-founded and aligns effectively with the goal of improving computational efficiency.
* **S2:** The Spatial Trimming Scheme offers a practical way to reduce redundant computation by concentrating on foreground regions, which is a sensible strategy.
* **S3:** The method demonstrates consistent improvements in both efficiency and generation quality across several benchmarks, supporting its practical value.

## Weaknesses and Suggestions for Improvement
* **W1:** The paper frequently describes the method as training-free (e.g., lines 5, 31, 51), but this characterization is somewhat misleading since a latent selector is trained to guide trajectory pruning. It would strengthen the paper to acknowledge this training component more transparently and clarify its role.
* **W2:** The explanation of the latent selector’s training process is occasionally unclear. For instance, lines 115–120 describe a training approach that the authors later state is not used, without sufficient explanation for its inclusion. Providing clarity here would help reduce ambiguity for readers.
* **W3:** Equation (1) is introduced but does not appear to be used in the final method, which makes its inclusion somewhat confusing. Meanwhile, Equation (2) is employed for training the latent selector, but key details—such as how the timestep \( t \) is sampled—are missing. Clarifying whether the selector is trained across all timesteps or at a fixed one would improve both the interpretability and reproducibility of the method.
* **W4:** Pruning all but one trajectory during inference may limit output diversity, as it commits early to a single generation path. Addressing how this approach impacts diversity, particularly in multi-modal generation tasks, would be valuable.
* **W5:** Some baseline results reported differ noticeably from those in the original papers, without any explanation or discussion. For example, in Table 2, the DiffSplat baseline shows PSNR, SSIM, and LPIPS scores of 16.20, 0.79, and 0.19, respectively, while the original DiffSplat paper reports 22.91, 0.892, and 0.107 for these metrics. Providing clarification on how baseline results were obtained and discussing these discrepancies would enhance the credibility of the comparisons.
* **W6:** The ablation study focuses primarily on variations in the latent selector architecture. It would be beneficial to also analyze the impact of individual trimming components—such as the Temporal Trimming Scheme, Progressive Mask Expansion Scheduler, Token Merging and Padding, and Post-denoising Correction—to better understand their contributions to overall performance.

---

> ### Author Rebuttal · Authors · 2025-07-31
>
> We sincerely appreciate the reviewer's thoughtful and detailed feedback. Your constructive suggestions are invaluable to improving our work, and we are encouraged by your recognition of our contributions. Below, we provide detailed responses to your concerns. As some of the weaknesses and questions are closely related, we merge their responses and address them together for coherence.
>
>
> **Weaknesses**:
>
> >W1: The paper frequently describes the method as training-free, but this characterization is somewhat misleading since a latent selector is trained to guide trajectory pruning.
>
> Our claim of "training-free" refers specifically to that we do not require any re-training or fine-tuning on diffusion models, meaning that weights of the off-the-shelf DiffSplat model remain completely unchanged in our framework. We acknowledge that our method introduces a lightweight latent selector, which is trained to adapt to the DiffSplat model and guide trajectory pruning.
> The selector is the only trainable component in our framework, and training it on our synthetic dataset takes less than 10 minutes, incurring only minimal training computational cost, as shown in training details in Appendix A. We will revise the paper to clarify this statement more rigorously in the next version.
>
>
> > W2, W3.1, Q2: The explanation of the latent selector's training process is occasionally unclear. For instance, lines 115–120 and Equation (1) describe a training approach that the authors later state is not used, without sufficient explanation for its inclusion.
>
>
> Thanks for your attention to our latent selector training process.
> We introduce the distillation objective in Equation (1) to set up the training goal, which is to align the selector's output with the Decoder-Renderer-Evaluator's score, which is the ground truth of the synthetic dataset.
> However, in the experiments of directly distiling the Decoder-Renderer-Evaluator joint model into the selector, we notice that the selector training is not stable and hard to converge. We attribute this training challenge to two reasons: 1) The variance of score distribution is very small, which makes it difficult for the selector to learn the slight score difference among multiple trajectories. 2) The lightweight selector is too weak to distill the complex knowledge in the Decoder-Renderer-Evaluator.
>
> Considering the above challenges on the direct distillation, we further propose a pairwise comparison training strategy to perform an indirect distillation, as shown in Line 143-158. The selector is changed to predict the relative performance among trajectories, which is more robust and stable to train. Compared to the direct distillation, the pairwise comparison training strategy is under the same offline distillation setting with the same training data and a similar training objective.
>
> We acknowledge that the explanation of the latent selector's training process is not clear in the submission. We appreciate your suggestion and will improve the method description to make the transition from direct model distillation to indirect pairwise comparison distillation smoother in the revised version.
>
>
>
> > W3.2, Q2: Equation (2) is employed for training the latent selector, but key details—such as how the timestep ( t ) is sampled—are missing. Clarifying whether the selector is trained across all timesteps or at a fixed one would improve both the interpretability and reproducibility of the method.
>
>
> We thank the reviewer for pointing out that Equation (2) does not mention the timestep. We clarify that Equation (2) reflects the final training objective used in our experiments, and the corresponding timestep setting is earlier described in Lines 144–145. Specifically, we use  $\mathbf{f}= \mathrm{CNN}(\mathbf{z}_t)$, the CNN embeddings of the intermediate latent features at timestep $t$,  as inputs to the MLP, and train the selector with the loss in Equation (2).
>
> We train the selector at a fixed timestep, and our ablation study in Figure 8 investigates the impact of different $t$ values. The performance gain plateaus after $t = 50\% T$, so we adopt this as the default timestep in our main experiments.
>
> We appreciate the reviewer's suggestion of training the selector across all timesteps. We agree that incorporating timestep as an additional input for the selector could turn to a more elegant and robust architecture without manually tuning the hyperparameter $t$. The timestep-aware selector might enable more flexible temporal trimming during inference. We plan to explore this advanced selector model in the future work.
>
>
> > W4: Pruning all but one trajectory during inference may limit output diversity, as it commits early to a single generation path. Addressing how this approach impacts diversity, particularly in multi-modal generation tasks, would be valuable.
>
> Thank you for your comment regarding output diversity. It is generally accepted that the diversity of diffusion models is primarily driven by the initial noise, i.e., different random seeds result in diverse generations. However, the quality of these diverse outputs can vary significantly.
>
> Our method is motivated by the observation that among multiple sampled trajectories, some yield substantially better results than others. As shown in the left subfigure of Figure 3, instead of choosing a random path, we employ a pairwise tournament selection strategy to identify and retain the trajectory with the most promising potential. This does not reduce the inherent diversity from random noise, but selects a higher-quality sample from the diverse pool. Thus, our approach balances maintaining diversity and improving output quality.
>
> Moreover, we can invert the usage of the selector to identify the "loser" trajectories, those predicted to have lower performance potential, offering a way to analyze the failure cases and lower-bound quality of the model's outputs.
> This dual capability of the well-trained selector enables more effective investigation of output diversity from both best-case and worst-case perspectives. We plan to explore this direction in future work.
>
>
> > W5, Q1: Some baseline results in Table 2 reported differ noticeably from those in the original papers, without any explanation or discussion.
>
> Thank you for pointing out the discrepancies. There are two main reasons for the differences between our reported baseline results and those in the original papers: 1）Data selection inconsistency: DiffSplat reports results on a randomly selected subset of 300 images from the GSO dataset, but does not disclose the its selection list. In our experiments, we randomly sample 300 GSO objects, matching the number used in DiffSplat, but it is not an identical subset to DiffSplat's. 2） Viewpoint rendering mismatch: Both DiffSplat and InstantMesh do not provide multi-view rendering camera parameters of the GSO evaluation dataset. Therefore, we follow the official GSO dataset's rendering protocol as an alternative, using four orthogonal views (front, back, left, right) rendered at a fixed elevation angle of 0°. This setting offers a standard and reproducible test set that is beneficial for future research. However, these four orthogonal views have lower overlap compared to six- or eight-view settings used in other works, making the reconstruction task more challenging. This explains why our reproduced baseline results are mismatched and lower than those reported in the original papers.
>
> We have included detailed settings about the GSO benchmark in Appendix A to ensure reproducibility. The results in Table 2 are conducted under these consistent settings for a fair comparison. We will provide additional explanation to discuss the differences in the GSO dataset settings in the revised version.
>
> > W6: The ablation study focuses primarily on variations in the latent selector architecture. It would be beneficial to also analyze the impact of individual trimming components—such as the Temporal Trimming Scheme, Progressive Mask Expansion Scheduler, Token Merging and Padding, and Post-denoising Correction—to better understand their contributions to overall performance.
>
>
> Thank you for the valuable suggestion to clarify the contribution of individual components. We provide an ablation study analyzing the contributions of Trajectory Reduction and Instance Masking in Figure 1 and Figure 2 of Appendix B. For the internal components of Instance Masking, we offer detailed discussion on their design and function in Section 3.3. Specifically, Instance Mask Detection and Token Merging & Padding are core components for spatial trimming and are indispensable to the overall pipeline. And we conducted additional ablations on the remaining two components: Progressive Mask Expansion Scheduler and Post-denoising Correction. The results in the table below show that removing these two components hurts the model output quality, demonstrating that both components contribute positively to the model's performance.
>
> Specifically, Progressive Mask Expansion Scheduler introduces a smooth and gradual transition from unmasked to masked regions, facilitating more stable and effective denoising. Post-denoising Correction removes Gaussian primitives located in transparent background regions, which are typically associated with [BG] tokens that are not optimally denoised, thereby reducing visual artifacts in the final rendered images.
> We will add visual comparisons of rendered outputs with and without Post-denoising Correction in the revised version to better illustrate its effect.
>
>
> | Method                                   | CLIP Sim.（%） | CLIP R-Precision（%） | ImageReward |
> | - | :-: | :-: | :-: |
> | TRIM                                     | 31.58          | 81.42                 | 0.12        |
> | w/o Progressive Mask Expansion Scheduler | 31.24          | 81.12                 | 0.03        |
> | w/o Post-denoising Correction            | 30.98          | 79.80                 | -0.28       |

---

> > ### Comment · Reviewer_CFPT · 2025-08-03
> > **Response to Authors's rebuttal**
> >
> > Thanks for the detailed response. Most of my concerns have been addressed, and your clarifications help.
> >
> > That said, I’m still not fully convinced about the claim that pruning early trajectories doesn’t hurt diversity because diversity mainly comes from initial noise. This point needs stronger evidence or analysis.
> >
> > If you can provide convincing experiments or details to back this up, I’d consider raising my score.
> >
> > Appreciate your effort and the thorough replies.

---

> > > ### Author Response · Authors · 2025-08-04
> > >
> > > Thank you for the further feedback. We're glad to hear that our clarifications have addressed most of your concerns.
> > >
> > > We also value your continued focus on the output diversity. To investigate this, we conduct additional experiments on T$^3$Bench's Single Object, measuring the diversity of generated outputs with or without trajectory reduction. We repeat the generation process 8 and 16 times, and report the average performance along with the standard deviation in the table below. The standard deviation is generally regarded as a key indicator for evaluating diversity.
> > >
> > >
> > >
> > > | Method  \ # repeat (Metric) |    8 (CLIP Sim.)     |    8 (ImageReward)    |  16 (CLIP Sim.)   |  16 (ImageReward)   |
> > > | --------------------------- | :------------------: | :-------------------: | :---------------: | :-----------------: |
> > > | w/o    Trajectory Reduction | 30.89   $\pm$   0.21 | -0.45   $\pm$   0.05  | 30.95  $\pm$ 0.16 |  -0.48  $\pm$ 0.04  |
> > > | w/ Trajectory Reduction     |  31.51  $\pm$ 0.14   | 0.12     $\pm$   0.04 | 31.53  $\pm$ 0.11 | 0.11     $\pm$ 0.03 |
> > >
> > >
> > > As shown in the table, we observe that the model's diversity is indeed slightly reduced when using trajectory reduction, with an increase in the average performance. We attribute this observation to the selector's ability to perform an early discarding of unpromising trajectories. By filtering out these less promising generation trajectories, the selector effectively shifts the output distribution of diversity toward a higher quality range. This means that we are sacrificing diversity on the low-quality outputs to improve the average quality of final outputs.
> > >
> > > Based on these experimental results, we would update our claim to state that initial noises primarily drive the diversity of the outputs, and our latent selector acts as a quality filter, enabling the effective selection from a pool of diverse candidates to high-quality outputs with a narrower variance.
> > >
> > > We very much appreciate your helpful suggestions, which encourage us to provide this deeper investigation. We will update our paper with these results and a more thorough analysis.

---

> > > > ### Comment · Reviewer_CFPT · 2025-08-05
> > > > **Response to the Authors**
> > > >
> > > > Thanks for the additional analysis. I appreciate the effort to investigate this further. However, I'm still unsure whether the reported standard deviation over CLIP Similarity and ImageReward scores accurately reflects the semantic or geometric diversity in a text-to-3D task.
> > > >
> > > > These metrics mostly capture alignment and quality, but not the spread of generated shapes, appearances, or poses. For instance, it's possible to have similar-looking outputs with varying CLIP scores due to small differences in texture or lighting.

---

> > > > > ### Author Response · Authors · 2025-08-06
> > > > >
> > > > > Thank you for your valuable suggestions. We understand that you are more concerned with the geometric diversity of generated outputs, and we agree that CLIP Similarity and ImageReward scores primarily focus on capturing semantic alignment and quality but do not fully represent the geometric information, such as shapes and poses.
> > > > >
> > > > >
> > > > > To address this, we conduct additional experiments to evaluate the geometric diversity of the outputs using **Chamfer Distance**. This metric is widely used for measuring the distance between two point clouds. Its output is a non-negative value, where a larger distance indicates a greater difference in shape and pose between the two point clouds.
> > > > >
> > > > >
> > > > > For the implementation details, we first extract the locations of Gaussian primitives from the generated 3D outputs, converting each object into a point cloud, as shown in the rightmost column of Figure 1.
> > > > > We then calculate the Chamfer Distance for every pair of point cloud outputs among the 4, 8, and 16 generated objects. The table below presents the averaged Chamfer Distance (non-normalized), along with the minimum and maximum values, for each setting. The upper bound (max value) of the Chamfer Distance is important to compare diversity, as it reflects the maximum pairwise difference among multiple outputs. Take the 8-output setting as an example, we calculate the Chamfer Distance for $\binom{8}{2}$ = 28 pairs of outputs and report the average, minimum, and maximum values.
> > > > >
> > > > >
> > > > > | Method  \ # repeat (Chamfer Distance) | 4 (avg, [min, max])  | 8 (avg, [min, max]) | 16 (avg, [min, max]) |
> > > > > | ------------------------------------- | :------------------: | :-----------------: | :------------------: |
> > > > > | w/o    Trajectory Reduction           | 2033,  [1133, 2938]  | 1840,  [952, 3060]  |  1791, [699, 4435]   |
> > > > > | w/ Trajectory Reduction               | 1995,   [1528, 2432] | 2042,  [1187, 2813] |  2310, [1187, 4079]  |
> > > > >
> > > > >
> > > > > For the geometric diversity, we notice that the range of Chamfer Distances is slightly narrowed and the upper bound is consistently lower when using trajectory reduction across all settings. These observations indicate that the trajectory reduction strategy slightly hurts the diversity of outputs and makes output distribution narrower.
> > > > >
> > > > >
> > > > >
> > > > > In summary, we have provided two kinds of experiments to explain the diversity from two aspects: 1) semantic diversity, measured by CLIP Similarity and ImageReward scores, and 2) geometric diversity, measured by Chamfer Distance.
> > > > > These two experiments consistently show that our trajectory reduction strategy slightly reduces the diversity of outputs, which is evidenced by the observed lower std in semantic metrics and narrower ranges of Chamfer Distance. This is well-aligned with our motivation to prune unpromising trajectories and shift the output distribution towards high-quality and narrow ranges.
> > > > >
> > > > > We value your question and try our best to address your concerns. We believe our analysis is comprehensive to back up our aforementioned claim. Please let us know if you have any further questions or would like to see additional experiments with other metrics beyond those we provided.
> > > > > We will include a detailed analysis of both the semantic and geometric diversity in the revised paper. Furthermore, since the rebuttal platform does not allow for image uploads, we plan to provide visual comparisons of the multiple outputs with and without trajectory reduction in the final version to further illustrate these findings.

---

### Official Review · Reviewer_9KEq · 2025-07-02

**Clarity:** 3
**Significance:** 3
**Originality:** 2
**Rating:** 3
**Confidence:** 4

**Summary:**

The authors propose TRIM (Trajectory Reduction and Instance Mask denoising), a training-free approach that incorporates both temporal and spatial trimming strategies. A lightweight selector model is introduced to evaluate latent Gaussian primitives derived from multiple sampled noises, enabling early trajectory reduction by selecting candidates with high-quality potential.

**Questions:**

1. how many trajectories do you have for TPIM? Does the Inference Step Scaling mean increasing denoising steps in inference?

**Ethical Concerns:**

["NO or VERY MINOR ethics concerns only"]

**Limitations:**

Yes

**Paper Formatting Concerns:**

/

**Quality:**

3

**Strengths And Weaknesses:**

strength

1. the authors demonstrate inference-time scaling in 3D diffusion models by trajectory pruning, showing that increasing the number of sampled trajectories during inference significantly improves the chance of producing high-quality results.
2. the authors conduct comprehensive ablation studies on computation, inference scaling, and selector.

weakness
1. the latent selector seems to rely heavily on the data synthesis and metrics score. What's the off-the-shelf image metric evaluator? Does it lead to overfitting and having to re-train the selector when changing the dataset?
2. The authors mention one of the significant improvements is the inference speed due to the Instance Masking (IM). Can the authors include the inference time comparison with other works?

---

> ### Author Rebuttal · Authors · 2025-07-31
>
> We sincerely appreciate the time and effort you dedicated to reviewing our work and your valuable comments. We provide the following clarifications and responses to address your concerns in detail.
>
> **Weaknesses**:
>
> >W1： The latent selector seems to rely heavily on the data synthesis and metrics score. What's the off-the-shelf image metric evaluator? Does it lead to overfitting and having to re-train the selector when changing the dataset?
>
> Thank you for the question. We provide the implementation details of selector training in Appendix A. Specifically, we construct a synthetic dataset using ChatGPT-generated prompts and 64 trajectories per prompt, and use CLIP score as the evaluation metric to supervise the selector. The well-trained selector is integrated into the TRIM pipeline to perform trajectory reduction (temporal trimming). It is noteworthy that this synthetic dataset is entirely independent of any downstream evaluation datasets, and no test sample is used during selector training, avoiding any test data leakage.
> In our text-to-3D generation experiments, the selector is trained using ChatGPT-generated prompts describing the single object, and it generalizes effectively to single-object with surrounding and multi-object generation tasks, as shown in Table 1. This indicates that the selector does not require re-training when the evaluation dataset changes.
>
> Regarding concerns about overfitting to the CLIP score, we empirically observe that the trajectories selected by our model also gain improvements under another ImageReward metric. This suggests that the selector, although trained with a single metric, generalizes well and provides improvements among other metrics.
>
>
>
> >W2. The authors mention one of the significant improvements is the inference speed due to the Instance Masking (IM). Can the authors include the inference time comparison with other works?
>
> Thank you for the suggestion. We have added inference time comparisons with two more baselines, LGM and InstantMesh, in the table below. Note that both LGM and InstantMesh adopt a different 3D generation pipeline that involves multi-view image diffusion followed by image-to-3D reconstruction. This pipeline uses a simpler U-Net architecture for image generation, resulting in faster runtimes than our Gaussian Splatting-based 3D diffusion model.
>
> Specifically, LGM uses 30 inference steps, while InstantMesh uses 75 steps, leading to InstantMesh having the longest runtime. In contrast, our TRIM uses the default 28 denoising steps but still achieves a significantly faster runtime than DiffSplat and InstantMesh, while being only slightly slower than LGM. We also provide the major performance PSNR results in the table below, which shows that TRIM achieves the best performance among all baselines and LGM performs not well on PSNR even though it has the fastest runtime.
> We will include the discussion of runtime differences between various 3D generation pipelines and baselines in the revised version.
>
> | Method      | Runtime (second) ↓ | PSNR ↑ |
> | ----------- | :----------------: | :---: |
> | LGM         |        4.93        | 14.90 |
> | InstantMesh |        9.78        | 15.53 |
> | DiffSplat   |        8.64        | 16.20 |
> | TRIM (Ours) |        5.24        | 16.78 |
>
>
>
> **Questions**:
>
> >Q1. How many trajectories do you have for TRIM? Does the Inference Step Scaling mean increasing denoising steps in inference?
>
> We discuss the number of trajectories used in TRIM in Section 4.4. The ablation experiments with 2,4,8 trajectories is shown in Figure 7, and the performance gain increases with more sampled trajectories. Based on this, we adopt 8 trajectories in our main experiments in Table 1 and Table 2.
>
> Regarding inference step scaling, as analyzed in Section 3.2, denoising N trajectories over T steps leads to a total of N×T denoising steps. Our temporal trimming scheme reduces denoising steps to N×T−(N−1)×t, where t is the early-stop trimming timestep. Thus, we save (N−1)×t steps overall.
>
> In implementation, we generate all N trajectories in parallel, so the total runtime is primarily determined by the longest (selected) trajectory, plus a small overhead from the pairwise tournament selection used in temporal trimming. As reported in Table 1 of Appendix A, this overhead is minimal and has little impact on overall runtime.

---

### Official Review · Reviewer_sj2g · 2025-07-02

**Clarity:** 3
**Significance:** 2
**Originality:** 3
**Rating:** 4
**Confidence:** 3

**Summary:**

This paper proposes a method to accelerate text-to-3D generation using 3D Gaussian diffusion by reducing inference time. Unlike existing state-of-the-art approaches that rely on lengthy denoising and post-processing, the authors introduce two key improvements. First, they generate synthetic data to train a selector model—comprising a CNN for feature extraction and an MLP for binary classification—that identifies the optimal latent trajectory early during inference. Second, they leverage the fact that image corners typically represent background, using these regions as reference to classify latent patches as foreground or background. This classification enables selective denoising, starting from outer regions and moving inward, thereby avoiding unnecessary computation. Their approach enhances both speed and quality in text-to-3D tasks using Gaussian diffusion models.

**Questions:**

To better evaluate the efficiency gains of the proposed method, please provide timing breakdowns for each added component—specifically the trajectory selection process and instance mask denoising steps.

**Ethical Concerns:**

["NO or VERY MINOR ethics concerns only"]

**Final Justification:**

The authors have addressed my original concerns in their thorough rebuttal. However, I find the diversity issue raised by Reviewer CFPT to be very valid. The authors responded to this point and clarified that it, in fact, it does reduce diversity, and I agree that this point should also be explicitly included in the final revision.

**Limitations:**

Yes

**Quality:**

3

**Strengths And Weaknesses:**

Strengths:

The paper is clearly written and easy to follow. It addresses a practically important and timely problem in 3D generation, inference efficiency and introduces a training-free, model-agnostic solution. The proposed method is both conceptually simple and effective. The authors support their claims with comprehensive experiments and ablation studies, demonstrating consistent improvements across multiple benchmarks.

Weaknesses:

The masking strategy relies on a corner-based attention heuristic, which assumes that objects are centered in the scene. This design limits generalizability to more diverse or complex layouts where foreground and background regions are not spatially separable.

Although the selector is described as lightweight, the paper does not provide a quantitative analysis of its memory or computational overhead.

The method, as explained in the paper, is not as effective on image-to-3D to 3D scenarios.

---

> ### Author Rebuttal · Authors · 2025-07-31
>
> We sincerely thank the reviewer for the thoughtful and constructive feedback. We are also pleased to see your recognition of the contributions of our work. We provide our clarifications in response to your concerns as follows. Due to the similarity between your provided Weakness 2 and Question 1, we address them together in the response to the weaknesses.
>
> **Weaknesses**:
>
> >W1: The masking strategy relies on a corner-based attention heuristic, which assumes that objects are centered in the scene. This design limits generalizability to more diverse or complex layouts where foreground and background regions are not spatially separable.
>
>
> Thank you for pointing out the potential limitation of our masking strategy based on the corner-based attention heuristic in complex layouts where foreground and background regions are not spatially separable.
>
>
> We would like to clarify that the center-object assumption is a widely adopted setting in the 3D object generation field. Specifically, 1) 3D datasets such as Objaverse and GSO (Google Scanned Objects) construct their coordinate systems with the target object centered, and both rendered and captured views are oriented towards the object center. 2) Many recent models, including LGM, InstantMesh, and DiffSplat, also train with object-centered renderings as input. Our approach follows these widely used settings for dataset preparation and model training to ensure comparability and relevance to standard benchmarks.
>
> We clarify that our method can also be applied in scenarios where foreground and background regions are not spatially separable. Notably, the "background" identified by our method refers to _invalid transparent regions_, rather than _semantically meaningful backgrounds_. Specifically, our approach detects regions where the Gaussian Splatting primitive's opacity attribute is zero or close to zero, which are effectively marked as transparent and uninformative. These regions are regarded as redundant primitives and then removed during the Transformer-based denoising process.
>
>
> We agree that generating complex layouts (e.g., multiple objects or objects with background context) is more challenging. To evaluate this, we have included evaluation results on such scenarios in Table 1 (T$^3$Bench's Single Object w/ Sur and Multiple Objects). Although performance drops slightly in these complex settings compared to single-object cases, our method still outperforms baselines, showing its effectiveness under complex layout settings. We appreciate your suggestion and will explore more robust strategies for complex layout generation in future work, especially for multi-object and background-aware scenes.
>
> >W2, Q1: Although the selector is described as lightweight, the paper does not provide a quantitative analysis of its memory or computational overhead.
>
> Thank you for the suggestion. We provide the computational cost of the selector in Table 1 of the appendix, as shown in the table below. Leveraging our latent selector introduces around 0.29 GB of additional GPU memory usage and 0.1 seconds of latency during inference. Overall, the marginal cost of the temporal selector is outweighed by the substantial efficiency improvements introduced by spatial trimming, making the combined trimming strategy both effective and resource-efficient.
>
>
> | Model         | FLOPs (T) ↓ | Mem. (GB) ↓ | Throughput (step/s) ↑ | Runtime (second) ↓ |
> | ------------- | :---------: | :---------: | :-------------------: | :----------------: |
> | SD-3.5-Medium |   195.68    |    33.26    |         13.18         |        8.64        |
> | + IM          |   165.60    |    32.85    |         18.09         |        5.16        |
> | + TR          |   110.07    |    33.55    |         13.18         |        8.74        |
> | + TRIM        |   106.31    |    33.13    |         18.09         |        5.24        |
>
> >W3: The method, as explained in the paper, is not as effective on image-to-3D to 3D scenarios.
>
>
>
> Thank you for the observation. Compared to text-to-3D generation, the image-to-3D task inherently exhibits lower output diversity, which reduces the gains brought by our temporal trimming. We attribute this to two main reasons: 1) Stronger supervision from input reference image: The reference image in image-to-3D provides a strong conditioning signal, making the sampled trajectories across different random seeds more consistent. This diminishes the benefit of pruning less useful trajectories since every trajectory may be similar. 2) Evaluation metric bias: The evaluation is based on accuracy compared to ground-truth views, which inherently penalizes plausible outputs that deviate from the ground-truth. Thus, the diversity encouraged by temporal trimming is under-reflected in the current reconstruction metric, since many generated details look plausible but are not considered correct under strict ground-truth comparisons.
>
> As a potential solution, we will incorporate diversity-aware metrics or human preference studies, like CLIP and aesthetic scorer, in future benchmarks to better evaluate the image-to-3D generation task.

---

> > ### Comment · Reviewer_sj2g · 2025-08-05
> >
> > Thank you for the detailed response. I appreciate the effort, and my initial concerns have been addressed. That said, I have also reviewed the comments from the other reviewers and find Reviewer CFPT's point regarding the effect of pruning on diversity very valid. I encourage you to incorporate a proper analysis of this aspect and revise the corresponding statement in the paper accordingly. Provided this is addressed, I am happy to maintain my current rating.

---

> > > ### Author Response · Authors · 2025-08-06
> > >
> > > Thanks for your response. We are pleased that our clarifications have addressed your initial concerns, and we appreciate your active engagement in the discussion with Reviewer CFPT regarding output diversity.
> > >
> > > Based on our two-round discussion with Reviewer CFPT, we provide a summary of our insights regarding output diversity. We have conducted additional experiments and detailed analysis on semantic diversity and geometric diversity via various metrics, including CLIP Similarity, ImageReward scores, and Chamfer Distance, as shown in the tables below.
> > > Here, CLIP Similarity and ImageReward scores primarily focus on capturing semantic alignment, while Chamfer Distance is used for measuring the distance between two point clouds, which is more sensitive to geometric diversity.
> > >
> > >
> > > | Method  \ # repeat (Metric) |    8 (CLIP Sim.)     |    8 (ImageReward)    |  16 (CLIP Sim.)   |  16 (ImageReward)   |
> > > | --------------------------- | :------------------: | :-------------------: | :---------------: | :-----------------: |
> > > | w/o    Trajectory Reduction | 30.89   $\pm$   0.21 | -0.45   $\pm$   0.05  | 30.95  $\pm$ 0.16 |  -0.48  $\pm$ 0.04  |
> > > | w/ Trajectory Reduction     |  31.51  $\pm$ 0.14   | 0.12     $\pm$   0.04 | 31.53  $\pm$ 0.11 | 0.11     $\pm$ 0.03 |
> > >
> > >
> > >
> > > | Method  \ # repeat (Chamfer Distance) | 4 (avg, [min, max])  | 8 (avg, [min, max]) | 16 (avg, [min, max]) |
> > > | ------------------------------------- | :------------------: | :-----------------: | :------------------: |
> > > | w/o    Trajectory Reduction           | 2033,  [1133, 2938]  | 1840,  [952, 3060]  |  1791, [699, 4435]   |
> > > | w/ Trajectory Reduction               | 1995,   [1528, 2432] | 2042,  [1187, 2813] |  2310, [1187, 4079]  |
> > >
> > >
> > > These two experiments show that our trajectory reduction strategy slightly reduces the diversity of outputs, resulting in a narrower output distribution.
> > > And these findings align with our core motivation: by pruning low-potential trajectories, we shift the output distribution toward high-quality results with a more reliable and narrower variance, which naturally reduces the overall diversity but significantly increases the average quality.
> > >
> > >
> > > We are confident that this analysis can address your and Reviewer CFPT's concerns. We will revise the corresponding statements, and fully include this detailed analysis and the new experimental results in the revised paper. We are grateful for your valuable feedback.

---

### Official Review · Reviewer_2U8q · 2025-07-03

**Clarity:** 3
**Significance:** 4
**Originality:** 3
**Rating:** 4
**Confidence:** 3

**Summary:**

This paper presents TRIM, a training-free method to improve the efficiency and scalability of 3D Gaussian diffusion models. TRIM reduces computation by selecting high-quality trajectories early and masking redundant background regions. It can be integrated with existing models without retraining, boosting both efficiency and quality in 3D generation tasks.

**Questions:**

1.Clarification of Definitions: Can you clarify Trajectory Scaling and Inference Step Scaling in the beginning of Ablation and Analysis?
2.Could you redraw Figure 6 to make it clearer, or consider splitting it into multiple figures for better clarity? This would help readers better understand the presented information.
3.More Details: Could you provide more detailed information on the experimental parameters, such as the optimal settings and the network architecture parameters?
4.Comprehensive Metrics: The experimental metrics used (CLIP, PSNR, SSIM, etc.) seem limited. Could you consider incorporating additional evaluation metrics, such as those in 3DGen-Bench or conducting a user study for more comprehensive results?
5.Comparison with Different Backbones: The paper mentions compatibility with various Transformer-based models, but lacks sufficient comparison across different backbones. Could you include experiments showing the impact of TRIM on various model architectures?
6.Qualitative Comparisons: The paper mainly compares with DiffSplat. Could you provide more qualitative comparisons with other state-of-the-art methods to better demonstrate TRIM’s performance and benefits?

**Ethical Concerns:**

["NO or VERY MINOR ethics concerns only"]

**Limitations:**

Yes

**Quality:**

3

**Strengths And Weaknesses:**

Strengths:
1.It proposes the TRIM framework, which effectively enhances the inference efficiency of 3D Gaussian diffusion models through temporal trajectory reduction and spatial instance mask denoising strategies.
2.The TRIM framework is plug-and-play, compatible with various Transformer-based 3D diffusion models without requiring retraining, demonstrating strong adaptability.
3.Extensive experiments on the DiffSplat backbone show that the model achieves approximately 45% reduction in computational resources (FLOPs) while improving generation quality, such as more detailed features in text-to-3D and image-to-3D tasks.
4.Ablation experiments thoroughly analyze the impact of inference step scaling and trajectory scaling on TRIM, verifying that sampling diverse trajectories significantly boosts generation quality compared to simply increasing denoising steps.

Weaknesses:
1.The writing lacks clarity in some areas. For example, the definitions of Trajectory Scaling and Inference Step Scaling should be provided earlier in the paper. Additionally, Figure 6 is difficult to understand. There is also a lack of implementation details and specific parameters used.
2.The experimental metrics are not sufficiently comprehensive. The paper uses CLIP similarity, CLIP R-Precision, ImageReward, PSNR, SSIM, and LPIPS to demonstrate the alignment between 3D objects and text/images, as well as human aesthetic preferences. However, these metrics are not exhaustive. A more comprehensive evaluation could be conducted by referring to the metrics used in 3DGen-Bench or by conducting a user study, which would provide a more thorough and nuanced assessment.
3.The paper claims to be compatible with various Transformer-based 3D diffusion models without requiring retraining, but it lacks sufficient comparisons to show how the performance changes when TRIM is applied to different backbones.
4.There is a lack of adequate qualitative comparisons; the paper only provides a direct comparison with DiffSplat, which is not enough for a thorough evaluation.

---

> ### Author Rebuttal · Authors · 2025-07-31
>
> We sincerely appreciate the reviewer's insightful and valuable comments. Below are our clarifications regarding your concerns. Due to the high similarity between your feedback in the weaknesses and questions, we merge some questions into the reply to the weaknesses and address them together.
>
> **Weaknesses**:
>
> > W1, Q1. The writing lacks clarity in some areas.
> > - The definitions of Trajectory Scaling and Inference Step Scaling should be provided earlier in the paper
>
> Thank you for the insightful suggestion. We agree that introducing Trajectory Scaling and Inference Step Scaling earlier in the paper would improve clarity and strengthen the motivation for our method. Our approach is driven by two key observations:
> 1）Increasing inference steps in 3D generation leads to diminishing gains, and even degradation when the number of steps becomes excessive.
> 2）Among multiple sampled trajectories, certain ones yield significantly better results than others.
>
> These findings demonstrate that the trajectory is a more effective axis to scale than the inference step, motivating us to propose Trajectory Scaling and the corresponding Trajectory Reduction methods. Therefore, moving the definition and analysis of these two kinds of scaling strategies to Sections 2 and 3, which are more closely associated with the core motivation, would help clarify the intuition behind our method and enhance the paper's clarity. We appreciate this helpful suggestion and will reorganize this motivation in the revised version.
>
>
> > W1, Q2. The writing lacks clarity in some areas.
> > - Figure 6 is difficult to understand.
>
> Thank you for your careful examination of our experimental results. We acknowledge that Figure 6 may be unclear in its current form.
> Figure 6 compares two different scaling strategies, Trajectory Scaling and Inference Step Scaling, each varying along different axes. Specifically, all experiments start from a baseline of 10 denoising steps and 1 trajectory (the leftmost bars in Figure 6):
>
> - Trajectory Scaling: We fix the number of denoising steps at 10 and increase the number of trajectories to {2, 4, 8}.
> - Inference Step Scaling: We fix the number of trajectories at 1 and increase the number of denoising steps to {20, 40, 80}.
>
> According to the above scaling settings, the total inference budgets (steps) are #steps $\times$ #trajectories, {10, 20, 40, 80}, shown as the x-axis of Figure 6.
> This comparison reveals the two observations discussed in our response to [W1, Q1]: 1) increased inference steps do not necessarily lead to better performance, and 2) sampling more trajectories offers more potential for selecting better outputs. These two observations show that the trajectory is a more effective axis to scale rather than the inference step. In Figure 6, the trajectory scaling strategy gradually outperforms the inference step scaling strategy and has more potential for better performance as the inference budgets increase.
>
> We acknowledge that the current legend and caption in Figure 6 are a bit confusing. In the revision, we will 1) clarify the axis labels and legends to match the original scaling strategies (TRIM $\rightarrow$ Trajectory Scaling, DiffSplat $\rightarrow$ Inference Step Scaling). 2) Rephrase the figure description in the paragraph for easier understanding.
> We appreciate your feedback.
>
>
>
> >W2: The experimental metrics are not sufficiently comprehensive. The paper uses CLIP similarity, CLIP R-Precision, ImageReward, PSNR, SSIM, and LPIPS to demonstrate the alignment between 3D objects and text/images, as well as human aesthetic preferences. However, these metrics are not exhaustive. A more comprehensive evaluation could be conducted by referring to the metrics used in 3DGen-Bench or by conducting a user study, which would provide a more thorough and nuanced assessment.
>
>
> We appreciate the reviewer's suggestion of using 3DGen-Bench and conducting a user study. Here, we use the CLIP-based 3DGen-Score model in 3DGen-Bench as an additional metric to enhance our evaluation. The 3DGen-Score model requires multi-view RGB images, normal maps, and a text prompt as input. The evaluation output contains five criteria: Geometry Plausibility, Geometry Details, Texture Quality, Geometry-Texture Coherence, and Prompt-Asset Alignment, each with different value scopes.
> We provide results on the T$^3$-Single text-to-3D benchmark in the table below, which is evaluated by the 3DGen-Score metric. The 3DGen-Score results show that TRIM achieves better performance than the baseline DiffSplat on all five criteria.
>
> | Method    | Geo. Plausibility | Geo. Details | Tex. Quality | Geo.-Tex. | Alignment |
> | --------- | :---------------: | :----------: | :----------: | :-------: | :-------: |
> | DiffSplat |      6462.91      |     8.14     |    13.43     | 15790.92  |  8544.38  |
> | TRIM      |      6567.58      |     9.25     |    13.47     | 16144.38  |  8585.02  |
>
> For a user study, we have included detailed visualization comparisons and utilized an aesthetic scorer to approximate human preference. As suggested, we plan to conduct a formal user study and include the results in future work.
>
>
> >W3, Q5: The paper claims to be compatible with various Transformer-based 3D diffusion models without requiring retraining, but it lacks sufficient comparisons to show how the performance changes when TRIM is applied to different backbones.
>
> We thank the reviewer for pointing out the importance of evaluating different backbones. Our proposed trajectory reduction strategy is designed to be applicable to most diffusion-based architectures, while the instance mask strategy specifically leverages the token structure of Transformer-based backbones. In our main experiments, we choose Stable Diffusion 3.5, the latest diffusion model in the Stable Diffusion series, as the backbone. To further demonstrate generalization, we also apply TRIM to PixArt-Sigma with a different diffusion backbone, and present the results on the  T$^3$-Single text-to-3D benchmark in the table below. These results show that TRIM improves the performance of DiffSplat with the backbone of PixArt-Sigma, demonstrating that TRIM consistently improves both the generation quality and efficiency across various Transformer-based backbones.
>
>
> | Method               | CLIP Sim.% ↑ | ImageReward Score ↑ | FLOPs (T) ↓ | Runtime (second) ↓ |
> | -------------------- | :----------: | :----------------: | :---------: | :----------------: |
> | SD-3.5-Medium        |    30.95     |       -0.49        |   195.68    |        8.64        |
> | SD-3.5-Medium + TRIM |    31.58     |        0.12        |   106.31    |        5.24        |
> | PixArt-Sigma         |    30.73     |       -0.30        |    25.18    |        2.76        |
> | PixArt-Sigma + TRIM  |    31.24     |       -0.13        |    14.16    |        2.19        |
>
>
> >W4, Q6: There is a lack of adequate qualitative comparisons; the paper only provides a direct comparison with DiffSplat, which is not enough for a thorough evaluation.
>
> We appreciate the reviewer's suggestion of providing more qualitative comparisons with other state-of-the-art methods. In the experiment section, we choose GVGEN, DIRECT-3D, LGM, InstantMesh, and DiffSplat as the baseline methods and show quantitative comparison results in Table 1 and Table 2. Since DiffSplat is the state-of-the-art method beyond the other three approaches, we present the qualitative comparisons with DiffSplat to highlight the improvement of TRIM. While the rebuttal does not allow us to insert the images, we will provide qualitative comparisons with more baselines, such as LGM and InstantMesh, in the updated submission.
>
> **Questions**:
>
> >Q3: More Details: Could you provide more detailed information on the experimental parameters, such as the optimal settings and the network architecture parameters?
>
>
> Thank you for your inquiry for more implementation details. We present our main experimental parameters in Appendix A.
> The ablation settings with custom parameter values are noted alongside each ablation experiment for clarity.
>
> For the 3D generation inference, we follow DiffSplat's default settings to ensure a fair comparison and reproducibility. Our 3D generation pipeline is built on the Stable-Diffusion-3.5-Medium backbone and uses the Flow Matching Euler ODE solver with 28 denoising steps and 8 sampled trajectories in the main experiments. The classifier-free guidance scale is set to 7. Input images are center-cropped and resized to 256×256 resolution.
>
> For the selector training, we adopt the Conv1-FC2 architecture for the selector, which achieves the best performance in the ablation study, shown as Table 3. We train the selector using the AdamW optimizer with a learning rate of 0.001, weight decay of 0.01, and a cosine decay scheduler. We set the training batch size to 64 and the epochs to 20.
>
> In addition, we detail the pairwise trajectory data synthesis procedure used to generate samples for training the selector in Appendix A.
> We appreciate your question and will provide more detailed implementation details in the revised version.

---

### Note · Authors · 2025-08-13

We appreciate all reviewers for their constructive feedback and valuable discussion, which have significantly improved the clarity and comprehensiveness of our submission.

**Strengths acknowledged by reviewers**


- **Contribution:** Our TRIM framework proposes temporal trajectory reduction and spatial instance mask denoising strategies to enhance the inference efficiency of 3D Gaussian diffusion models effectively.
- **Analysis:** Comprehensive evaluation and ablation studies are conducted to analyze computation, inference scaling, selector training, and output performance.


**In-depth discussions**

- **Clarity of Presentation:**  We provide detailed explanations for selector training,  temporal and spatial trimming strategies, analysis of inference scaling, the revised claim of "training-free", and implementation details with parameter settings.


- **Experiment:** We incorporate the (1) additional metrics 3DGen-Score and Chamfer Distance, (2) another backbone PixArt-Sigma, (3) an additional runtime comparison with baselines LGM and InstantMesh, (4) additional ablation studies to clarify the contribution of individual components, and (5) experiments to analyze the impact of trajectory pruning on output diversity.

- **Diversity Analysis:** We appreciate the insight from Reviewer sj2g and CFPT regarding the impact of trajectory pruning on output diversity. This keeps pushing us to conduct deeper experiments and analyze both semantic diversity and geometric diversity. The results show that our method slightly reduces overall diversity but significantly improves the average quality by filtering out low-potential trajectories. This finding confirms our core motivation, which is to generate high-quality outputs with a narrower variance.

**Reviewer updates**

- We appreciate Reviewers sj2g and CFPT's expertise and discussion during the rebuttal, and we are pleased to be informed that their concerns have been addressed. Reviewer sj2g maintains the score, and Reviewer CFPT considers raising the score.

- Although Reviewer 2U8q and 9KEq didn't participate in the rebuttal, we still appreciate their valuable suggestions and believe our replies provide a comprehensive and detailed explanation to address their concerns.

We will include these suggestions and additional results in the revised paper to present a clearer and more comprehensive insight into the field.

---

### Decision · Program_Chairs · 2025-09-17

**Decision:**

Accept (poster)

**Comment:**

The submission studies 3D Gaussian diffusion models. Specifically, the submission argues that the large number of Gaussians leads to a slow denoising, i.e., slow generation. To address, the authors incorporate temporal and spatial trimming strategies. Initially, the reviewers raised concerns among others about clarity (2U8q, CFPT), missing comparison/incomprehensive evaluation (2U8q, sj2g, 9KEq, CFPT), unjustified/misleading claims (2U8q, CFPT), generalizability (sj2g), effectiveness (sj2g), differences to numbers reported in prior work (CFPT). The rebuttal was able to answer many of these concerns. A remaining concerns was the reduction in diversity due to pruning. The author discussion showed that the proposed approach indeed reduces diversity. Two reviewers strongly encouraged the authors to include this discussion in the revised submission.